# Tissue-location-specific transcription programs drive tumor dependencies in colon cancer

Lijing Yang[1,2,6], Lei Tu[3,6], Shilpa Bisht [1], Yiqing Mao[1], Daniel Petkovich [1], Sara-Jayne Thursby[1], Jinxiao Liang[1], Nibedita Patel[1], Ray-Whay Chiu Yen[1], Tina Largent[1], Cynthia Zahnow[1], Malcolm Brock[1], Kathy Gabrielson[4], Kevan J. Salimian[5], Stephen B. Baylin [1] & Hariharan Easwaran [1] ✉

Cancers of the same tissue-type but in anatomically distinct locations exhibit different molecular dependencies for tumorigenesis. Proximal and distal colon cancers exemplify such characteristics, with $BRAF^{V600E}$ predominantly occurring in proximal colon cancers along with increased DNA methylation phenotype. Using mouse colon organoids, here we show that proximal and distal colon stem cells have distinct transcriptional programs that regulate stemness and differentiation. We identify that the homeobox transcription factor, CDX2, which is silenced by DNA methylation in proximal colon cancers, is a key mediator of the differential transcriptional programs. Cdx2-mediated proximal colon-specific transcriptional program concurrently is tumor suppressive, and Cdx2 loss sufficiently creates permissive state for $BRAF^{V600E}$-driven transformation. Human proximal colon cancers with CDX2 downregulation showed similar transcriptional program as in mouse proximal organoids with Cdx2 loss. Developmental transcription factors, such as CDX2, are thus critical in maintaining tissue-location specific transcriptional programs that create tissue-type origin specific dependencies for tumor development.

Cancers occurring in different anatomical sites within same tissue type and organ can exhibit stark differences in molecular, pathological, and clinical features. Colon cancers exemplify such features wherein there is an increased likelihood of exhibiting the CpG-island methylator phenotype high (CIMP-high or CIMP-H), microsatellite instability high (MSI-high) and having BRAF mutations to dominate in the cancers occurring in the proximal colon[1,2]. Distal colon tends to have CIMP-Low, microsatellite stable (MSS), APC-mutant cancers, with an increased likelihood of KRAS mutations[3,4]. The mutually exclusive BRAF and KRAS mutations provide alternate means of activating the MEK-

ERK signaling pathway and function as critical cancer driver mutations in colon cancers. Genome-wide association studies have shown further differences in genetic risk factors in proximal and distal colon cancer[5]. Developmentally, the proximal colon derives from the midgut and extends from the cecum to the proximal two-thirds of the transverse colon, while distal colon derives from the hindgut and extends from the distal third of the transverse colon to the sigmoid colon[6]. Anatomically, the proximal and distal colons have distinct blood supplies, with the superior and inferior mesenteric arteries supplying to proximal and distal colon respectively[6]. And microenvironmentally,

[1]CRB1, Department of Oncology and The Sidney Kimmel Comprehensive Cancer Center at Johns Hopkins, The Johns Hopkins University School of Medicine, Room 530, Baltimore, MD 21287, USA. [2]Department of Radiation and Medical Oncology, Zhongnan Hospital of Wuhan University, Wuhan, Hubei 430071, PR China. [3]Division of Gastroenterology, Union Hospital, Tongji Medical College, Huazhong University of Science and Technology, Wuhan 430022, China. [4]Department of Comparative Medicine, Johns Hopkins Medical Institutions, 863 Broadway Research Building, 733 N. Broadway, Baltimore, MD 21205-2196, USA. [5]Department of Pathology, The Johns Hopkins University School of Medicine, Baltimore, MD 21287, USA. [6]These authors contributed equally: Lijing Yang, Lei Tu. ✉e-mail: heaswar1@jhmi.edu

proximal and distal colon have differences in microbiome[7,8] and exposures to chemicals in the food as they pass along the axis of the colon[9]. What defines the mechanisms through which cancer cell-of-origin at sites along the proximal to distal (rostrocaudal) colon axis evolve different molecular dependencies for tumor development is not yet clarified. Dissecting this biology is important to understanding the basis for key differences in the molecular genetics and clinical features of these cancers, and in general for understanding why cancers arising within highly similar tissues may exhibit distinct features.

In relation to the molecular dependencies for cancer development, tumorigenesis involves profound alterations to the expression or activity of key transcription factors (TFs) in association with cancer driver mutations[10–12]. These TFs constitute combinations of cell and tissue-type lineage-specific transcription factors that normally maintain transcriptional programs important in development and differentiation[13–15]. The cell-type-specific transcriptional programs are important for maintaining cellular identities, and their alterations play important roles in mediating effects of cancer driver mutations[11]. The fact that the developmental and anatomical context of tissues are associated with different molecular dependencies for tumor initiation[16,17] points to potential involvement of developmentally controlled transcriptional states and epigenetic programs in mediating effects of cancer driver mutations and driving tumor phenotypes.

A key feature of the proximal colon cancers with $BRAF^{V600E}$ mutation is their increased likelihood to have the CIMP-H phenotype, leading to epigenetic silencing of critical tumor suppressors, including the key cell cycle checkpoint regulator $CDKN2A$, the Wnt-pathway regulators such as the $SFRPs$ and $SOX17$, and developmental regulators including the key intestinal lineage determining homeobox TF, $CDX2$[18–20]. Work from our lab has shown that loss of these key epigenetically silenced genes predispose colon stem cells to undergo tumorigenesis by $BRAF^{V600E}$ mutation[19]. However, considering the epigenetic variations in the CIMP phenotype in proximal and distal colon cancers, the impact of inactivation of these genes in proximal and distal colon cancer development is not known.

Here we show that the TF-mediated differential transcriptional states in proximal and distal colon stem cells are critical in driving differential dependencies for tumorigenesis in proximal and distal colon. Using mouse-derived organoids from normal proximal and distal colon, our studies show that loss of the homeobox transcription factor $CDX2$, which is important in lineage determination of colon epithelium during development[21–28], is critical in facilitating the dependency of the $BRAF^{V600E}$ mutation in driving proximal colon cancers. Mechanistic studies reveal important roles of Cdx2 in regulating stem and differentiated cell states specifically in proximal colon epithelial cells. The studies reveal that the transcriptional programs driven by key lineage specification TFs, such as CDX2, are critical determinants of tissue-site-specific susceptibility to colon cancer development by cancer driver mutations. These studies have implications in understanding why very closely related tissues exhibit different molecular genetic features, as well as in understanding the parameters that lead to driving tumorigenesis by these mutations

## Results
### Abnormal activation of Wnt pathway facilitates Wnt-factor-independent growth in both proximal and distal colon organoids
We started our studies by exploring basal differences in stem cell activity and tumorigenic potential in proximal and distal colon stem cells in the context of strong Wnt-activating mutations. To this end, we derived proximal and distal colon organoids from two months old mice carrying heterozygous Cre-activable $BRAF^{V600E}$ (termed Braf$^{+/LSL}$) transgene at the endogenous $Braf$ locus and Cre-activable gene encoding red fluorescence marker protein, TdTom (termed TdTom$^{+/}$ $^{LSL}$), to mark cells that have undergone Cre-mediated recombination of

the above alleles (Supplementary Fig. 1A). Cre-mediated recombination generates a mouse-human hybrid form of the V600E variant allele, which we label as $BRAF^{V600E}$ as per previous nomenclature[19,29]. Both the wild-type proximal and distal organoids have typical organoid morphology with cystic structures and "crypt" like, stem cell buds[30]. In general, the proximal organoids tended to grow into larger structures with small buds while the distal organoids were smaller with more budded protrusions (Supplementary Fig. 1B–E).

As Wnt-pathway activation is a key dependency for colon cancers, which in the majority cases occurs via inactivating mutations in $APC$, and in rare cases by activating mutations in $CTNNB1$ (β-catenin)[31,32], we studied Wnt-independent growth properties of the proximal and distal organoids by introducing these mutations using CRISPR (clustered regularly interspaced short palindromic repeats)-Cas9 and single-guide RNAs (sgRNAs) targeting these genes (Fig. 1a). Organoid clones with integrated lentiviral constructs expressing Cas9-CRISPR/Apc-sgRNA or Ctnnb1-sgRNA were obtained after a two-week drug selection. Both Apc-sgRNA and Ctnnb1-sgRNA organoid clones exhibited ability to grow in medium lacking stem cell niche factors required for organoid growth (Wnt3a, EGF, Noggin, and R-Spondin-1), termed WENR-minus medium. Clones able to grow in WENR-minus medium harbored truncating mutations in $Apc$ or mutations affecting phosphorylation sites in $Ctnnb1$ (β-catenin gene) (Supplementary Fig. 1F, G).

As a key property of colon stem cells is the ability of single stem cells to independently regenerate the organoid structures[30,33], the organoid regeneration capacity of these genetically edited proximal and distal colon organoids in Wnt-independent growth factor conditions was evaluated. As expected, single cells derived from organoids harboring the selected mutations in $Apc$ or $Ctnnb1$ exhibited constitutive Wnt activation, rendering these organoids to grow indefinitely in medium lacking stem cell niche factors (WENR-minus) upon subsequent passages. In contrast, the control (Scramble-sgRNA) organoids could be cultured only in WENR-plus medium, containing all niche factors (Fig. 1b). Organoids with $Apc$ or $Ctnnb1$ mutations exhibited the typical spheroidal structures lacking the budded patterns indicating loss of differentiated cells and increased stem cell properties (Supplementary Fig. 1D, E)[33]. Thus, both proximal and distal organoids acquire Wnt-independent growth properties upon $Apc$ or $Ctnnb1$ mutations, in concordance with expectations from previous studies that constitutive Wnt activation due to these mutations imparts stem cell renewal and growth in the absence of Wnt-pathway activating ligands in the growth medium[33,34].

### Differences in $BRAF^{V600E}$-driven tumors in proximal and distal colon stem cells with abnormal Wnt activation
As introduced earlier, colon cancers occurring in the proximal and distal colon tend to exhibit distinctive genetic and epigenetic alterations, with those in proximal colon cancers more frequently harboring $BRAF^{V600E}$ mutations and being mutually exclusive of $APC$ mutations[1,3]. It is not known whether the $BRAF^{V600E}$ mutation has differential effects on transformation of proximal and distal colon stem cells, and especially how this is modulated by the key Wnt-activating $APC$ mutations. To address this, we activated the $BRAF^{V600E}$ mutation in the proximal and distal colon organoids by introducing lentiviral-expressed Cre recombinase, and monitored successful Cre-mediated recombination using the red fluorescent $TdTom$ reporter as in our previous work[20] (Fig. 1b). In terms of Wnt-independent growth ability, both proximal and distal organoids harboring $Apc$ or $Ctnnb1$ mutation with activated $BRAF^{V600E}$ mutation were able to grow equally well in WENR-minus and plus medium. However, organoids with only $BRAF^{V600E}$ (wild type for $Apc$ or $Ctnnb1$) did not exhibit Wnt-independent growth but grew proficiently in medium containing all colon stem cell niche factors (Fig. 1b). Thus, constitutive Wnt activation by $Apc$ loss or $Ctnnb1$ gain-of-function mutations is a strong driver of growth and organoid

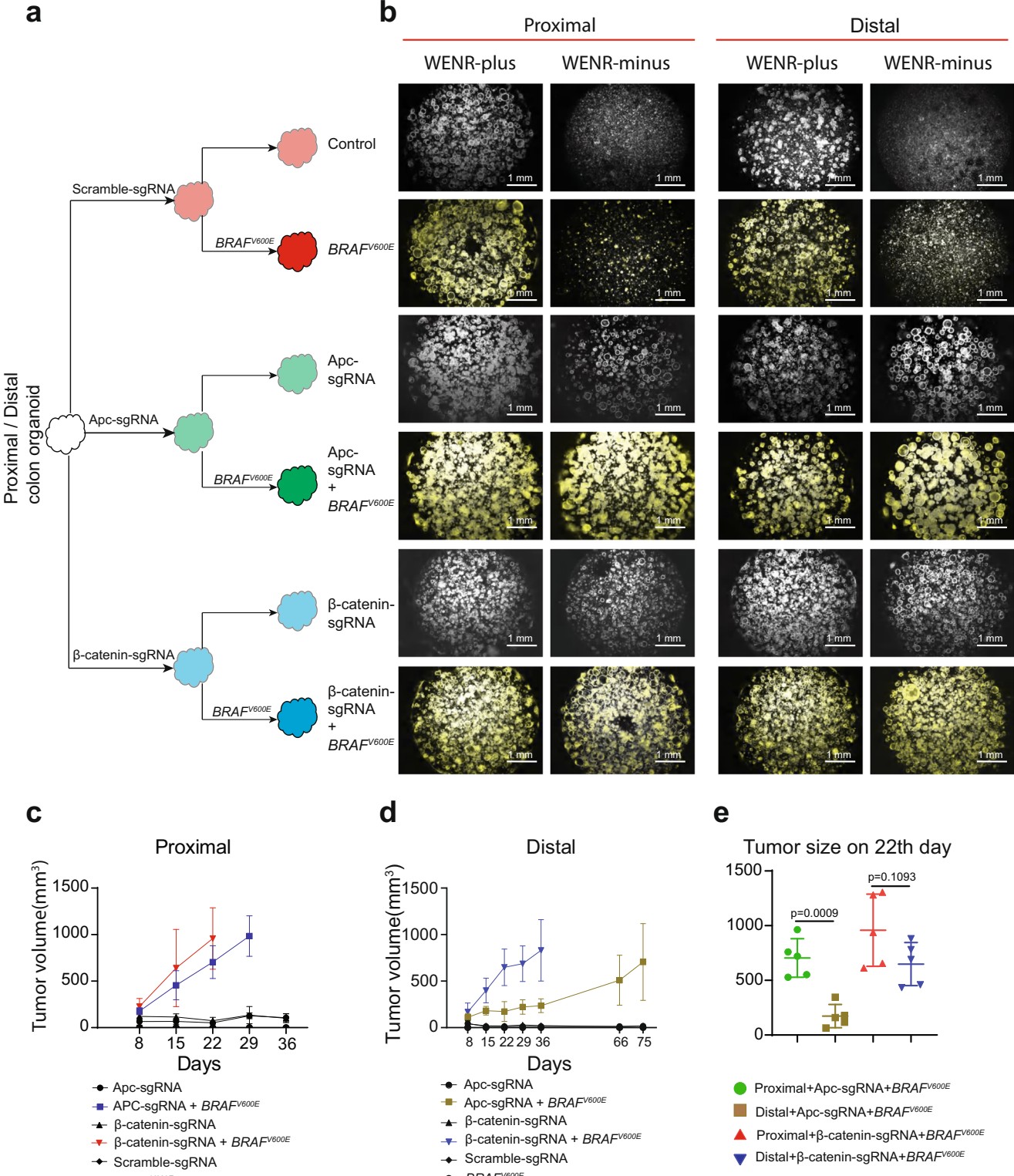

**Fig. 1 | Abnormal Wnt pathway activation facilitates $BRAF^{V600E}$ mutation-driven tumorigenesis in proximal and distal organoids derived from mouse colon.**
**a** Schematic representation of gene editing of proximal or distal colon organoid using CRISPR-Cas9 targeting *Apc* or N-terminal of *Ctnnb1* (β-catenin) gene, followed by induction of $BRAF^{V600E}$ by Cre recombinase. **b** Representative composite images (phase-contrast and red fluorescent) of Wnt-factor independency analysis for proximal/distal colon organoid with/without abnormal Wnt-pathway activation (Apc-sgRNA, β-catenin-sgRNA group), and with *Braf* wild type or $BRAF^{V600E}$ (growth assay data are representative of two biological replicates). **c, d** Subcutaneously grafted organoid-derived tumor growth curve of proximal (**c**)/distal (**d**) colon organoid with/without constitutive Wnt-pathway activation by mutant *Apc* or *Ctnnb1* (β-catenin) (APC/ β-catenin-sgRNA) and with/without Cre-mediated $BRAF^{V600E}$ activation injected subcutaneously into NSG mice. Error bars indicate means ± SD (*n* = 5 mice, right flank each mouse, one biological organoid replicate injected in the mice for assays). Source data are provided in FigurePlotsSourceData.xlsx. **e** Tumor volume formed by proximal versus distal colon organoids at day 22. Error bars indicate ± SD (*n* = 5 mice, right flank each mouse, one biological organoid replicate injected in the mice for assays). Two-sided Welch's *t*-test. Source data are provided in FigurePlotsSourceData.xlsx.

morphology in both proximal and distal organoids which is not affected by the $BRAF^{V600E}$ mutation.

Tumorigenesis assays by subcutaneous transplantation of organoids in NSG mice showed that cells harboring only the $BRAF^{V600E}$ mutation or $Apc$ or $Ctnnb1$ mutations were not able to form tumors indicating the requirement of the combination of MEK-ERK activation by $BRAF^{V600E}$ along with Wnt activation for tumorigenesis (Fig. 1c–e). This is consistent with previous studies showing that $Apc$ loss alone in organoids does not form tumors upon subcutaneous engraftment[34]. When compared with distal organoids, the proximal organoids with $Apc$ mutation and $BRAF^{V600E}$ showed more rapid tumor growth resulting in larger tumor volumes (Fig. 1e). In H & E staining, tumor cells derived from proximal and distal colon organoids both formed abundant cystic structures (Supplementary Fig. 2A, B). But the tumor cells derived from the proximal colon organoids were small, with a high nuclear-to-cytoplasmic ratio, and without vesicular structures. Tumor cells derived from the distal colon organoids were bigger with a low nuclear-to-cytoplasmic ratio, large vesicular structures at the apical surface, and basally positioned nuclei. Further, $BRAF^{V600E}$-driven tumors in the context of mutant $Apc$ from distal organoids presented more clusters of differentiated cell morphologies compared to those from proximal organoids in H&E staining, and a tendency to have increase in Krt20-positive cells (Supplementary Fig. 2A, B). The above results imply that while both proximal and distal organoids with constitutive Wnt-pathway activating $Apc$ or $Ctnnb1$ mutations and $BRAF^{V600E}$ mutation undergo transformation and ability to form tumors, they have a tendency to exhibit differences in stem cell and differentiation potential during tumorigenesis.

## Loss of Cdx2 gene imparts transient Wnt-independent growth specifically in proximal colon organoids

In view of all the above results, and the fact that $BRAF$ mutations occur at higher incidence in the context of proximal colon cancers characterized by the CIMP-H phenotype[1], we focused our studies to determine the impact of loss of genes subject to frequent silencing by promoter hypermethylation in proximal CIMP-H cancers. In our previous work we have shown that simultaneous inactivation of key genes that get typically silenced by promoter hypermethylation in human colon cancers, viz. $CDX2$, $CDKN2a$, $SFRP4$, and $SOX17$, drives $BRAF^{V600E}$-induced tumor development in the proximal colon-derived organoids from mouse[20]. These genes are all subject to frequent loss by promoter hypermethylation and rarely by mutations in human proximal colon cancers. However, it is not known if inactivation of these genes plays differential roles in proximal and distal colon cancer development. Among these genes, $CDX2$ is a homeobox transcription factor that is important for lineage and segment specification of the small intestine during development of the GI tract[21–28], which in accordance with previous studies shows a gradient of high to low expression along the proximal to distal (rostrocaudal) colon axis in adult mice (Supplementary Fig. 6E) and human[35]. Sfrp4, Sox17, and Cdkn2a have similar expression in proximal and distal colon, with Cdkn2a showing upregulation upon $BRAF^{V600E}$ induction (Supplementary Fig. 6F). We thus addressed the role of $Cdx2$ and the other key epigenetically silenced genes, i.e., $Cdkn2a$, $Sfrp4$, and $Sox17$, in facilitating tumorigenesis driven by $BRAF^{V600E}$ in the proximal and distal colon organoids.

We generated proximal and distal colon organoids harboring gene mutations (indels) using CRISPR-Cas9 and sgRNA guides individually targeting these genes, termed Cdx2-sgRNA, Cdkn2a-sgRNA, Sfrp4-sgRNA, and Sox17-sgRNA (Fig. 2a, Supplementary Fig. 3A–D, Supplementary Table 1). Single cells derived from these organoids were evaluated for Wnt-independent growth ability. Cells harboring individual mutation of these four genes were able to form new organoid structures in normal medium indicating preservation of organoid regenerative capacity in the presence of exogenous Wnt factors (WENR-plus). In contrast, only single cells generated from proximal

organoids with the $Cdx2$ mutation regenerated organoid structures when challenged to grow in WENR-minus medium (Fig. 2b). Neither the proximal nor the distal organoids harboring Cdkn2a-sgRNA, Sfrp4-sgRNA, and Sox17-sgRNA survived in WENR-minus medium (Fig. 2b). $Cdx2$ loss in proximal, but not distal, organoids altered overall growth morphology independent of exogenous Wnt signaling, observed as spheroidal-like oblong morphology, when grown in WENR-plus medium (Supplementary Fig. 3E, F). This morphology differed from that of large cystic, spheroidal organoids due to $Apc$ and $Ctnnb1$ mutation (Supplementary Fig. 1D).

We explored whether $Cdx2$ mutation imparts complete Wnt-independency similar to that of mutations in $Apc$ or $Ctnnb1$. As mentioned earlier, Apc-sgRNA or Ctnnb1-sgRNA organoids exhibit complete and autonomous Wnt-pathway activation promoting indefinite stemness, wherein enzymatically separated single cells from these organoids can be indefinitely cultured in WENR-minus medium at successive passages. In stark contrast to the Apc-sgRNA or Ctnnb1-sgRNA organoids, Cdx2 loss resulted only in transient stemness and organoid regeneration properties. Single cells derived from Cdx2-sgRNA organoids surviving in the WENR-minus medium during the first week after transfer from WENR-plus medium did not form new organoids when challenged to again grow in WENR-minus medium for a second passage (Fig. 2b, bottom panel). These same Cdx2-sgRNA single cells formed organoids in WENR-plus medium at subsequent passages, thus indicating that exogenous Wnt-ligands in the medium is required for organoid formation (Fig. 2b). Interestingly, the Cdx2-sgRNA cells that transiently survive in WENR-minus medium, continue to grow into larger and fused structures when not re-passaged any further, suggesting maintenance of differentiated cells without further proliferation.

The above results suggest that Cdx2 plays differential roles in proximal and distal organoids. Its loss specifically in proximal organoids allows transient Wnt-independent growth without activating the canonical stem cell activity, which is typically activated in colon epithelial cells upon loss of $Apc$ or $Ctnnb1$ functions.

## $Cdx2$ loss drives transient Wnt-independent growth specifically in proximal organoids by altering epithelial cell differentiation state

Previous studies have shown that Cdx2 is an important regulator of differentiation in colonic epithelium[36]. We thus compared the impact of $Cdx2$ loss with that of $Apc$ loss on the stem cell and differentiation states by monitoring expression of key intestinal stem cell (ISC) (Ephb2, Lgr5, Ascl2) and differentiation (Fabp2, Krt20, Car1, and Muc2) markers (Fig. 2c, Supplement Table 2)[37]. Organoids carrying Apc or Cdx2-sgRNA were cultured in WENR-plus medium for 2 days. Subsequently, they were cultured in WENR-plus or WENR-minus medium for 3 days before assessing their growth and monitoring the expression of stem cell and differentiation markers. We chose 3 days for the qPCR (and RNA-seq) analyses as beyond this time point the control (reference) organoids in WENR-minus start differentiating and disintegrating. As expected, $Apc$ loss resulted in upregulation of the stem cell markers in WENR-plus medium, but importantly also in the organoids growing in WENR-minus medium. In contrast, although ISC markers increased in Cdx2-sgRNA organoids in relation to control (Scramble-sgRNA) organoids, most significant being EphB2, this phenomenon is dependent on exogenous Wnt activation as these ISC markers were starkly downregulated in Cdx2-sgRNA organoids grown in WENR-minus medium (Fig. 2c). Thus, like that in the wild-type organoids, activation of stem cell pathway in the Cdx2-mutant organoids depends on exogenous Wnt-ligands. Differentiation states of Cdx2-sgRNA organoids were significantly affected in comparison to wild-type organoids, with significant downregulation of $Fabp2$, $Car1$, and $Muc2$, while an upregulation in the key epithelial cell marker, $Krt20$ (Fig. 2c). In contrast, Apc-sgRNA organoids showed downregulation of all differentiation markers, including $Krt20$, consistent with its increased

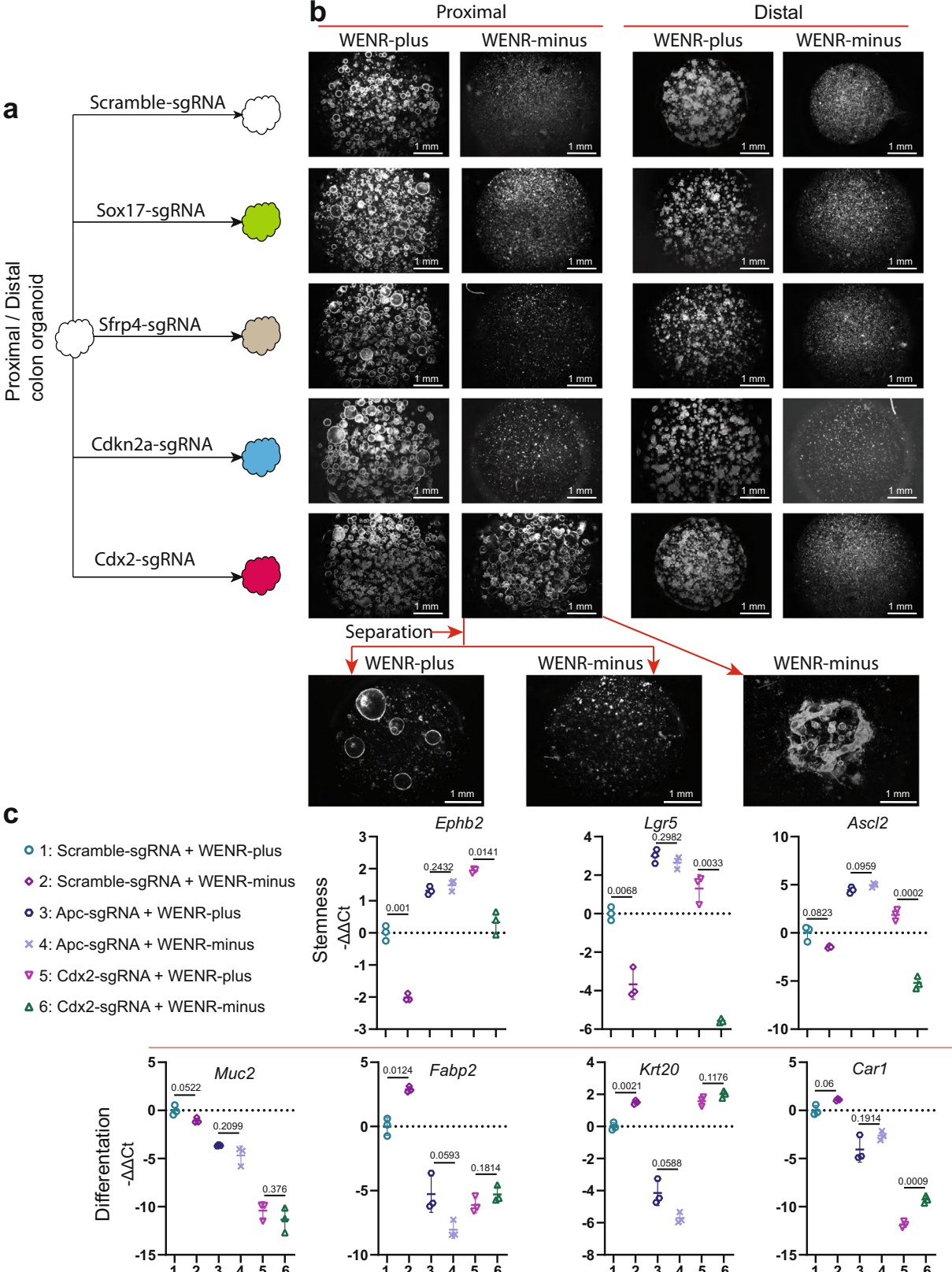

**Fig. 2 | Loss of *Cdx2* gene imparts transient Wnt-independent growth specifically to proximal colon organoids. a** Schematic representation showing proximal or distal colon organoids transduced with lentivirus containing Scramble-sgRNA, Sox17-sgRNA, Sfrp4-sgRNA, Cdkn2a-sgRNA, or Cdx2-sgRNA to individually target these genes using CRISPR-Cas9. **b** Representative images showing Wnt-factor-independent growth of proximal and distal colon organoids transduced with lentivirus expressing the Scramble-sgRNA, Sox17-sgRNA, Sfrp4-sgRNA, Cdkn2a-sgRNA, or Cdx2-sgRNA targeting these genes separately using CRISPR-Cas9 technology (growth assay data are representative of two biological replicates). **c** Quantitative real-time PCR analysis of stemness (*Ephb2*, *Lgr5*, and *Ascl2*) or differentiation (*Muc2*, *Fabp2*, *Krt20*, and *Car1*) markers for proximal colon organoids with Cdx2-sgRNA or Apc-sgRNA and cultured in WENR-plus or -minus medium. *n* = 3 biological replicates. Error bars indicate means ± SD. Two-sided Welch's *t*-test. Source data are provided in FigurePlotsSourceData.xlsx.

stem cell state. Thus, while *Apc* loss resulted in sustained activation of stemness, *Cdx2* loss mainly resulted in altered differentiation states with partial increases in Wnt-pathway factors. It is important to note that Cdx2 deficiency results in upregulation of the key stemness genes only in the presence of external Wnt-stimulation (WENR-plus). As shown in Fig. 2c, in the absence of Wnt-ligands (WENR-minus), Lgr5 and Ascl2 are downregulated in both presence (Scramble-sgRNA + WENR-minus) or absence (Cdx2-sgRNA + WENR-minus) of Cdx2. EphB2, which is another stem cell marker known to be high in intestinal stem cells, is not altered much in Cdx2-sgRNA organoids transferred to WENR-minus medium (condition-6, Fig. 2c), indicating that these stem cell markers follow different dynamics of expression in the 3-day period post transfer of the organoids to WENR-minus medium. Further, as shown in the later sections, Cdx2 loss does not directly activate the Wnt/β-Catenin pathway (Supplementary Fig. 5G). These data thus indicate that Cdx2 does not directly suppress Wnt-target genes, which may explain why Cdx2 loss results in only the transient Wnt-independent growth properties shown in the previous section.

Further, H&E and immunohistochemistry (IHC) analyses highlight partial activation of the Wnt pathway in the survival of Cdx2-deficient organoids in WENR-minus medium compared to Apc-deficient organoids (Supplementary Fig. 2C, D). Control (Scramble-sgRNA) organoids in WENR-minus medium showed more frequent disintegration of organoid structures, with the cells displaying increased vacuolar-like structures. Both Apc-sgRNA and Cdx2-sgRNA organoids maintained structural integrity in both WENR-plus and minus conditions. Cdx2-sgRNA proximal organoids cultured in WENR-minus medium exhibited a thin epithelial layer and reduced size of the epithelial cells, compared to those grown in WENR-plus medium. No such differences were observed for the Apc-sgRNA proximal organoids (Supplementary Fig. 2C). IHC for activated β-catenin indicated that Cdx2 loss did not result in complete activation of canonical Wnt pathway as highlighted by the distribution of β-catenin at the cell membrane and junction between colon cells in the Cdx2-sgRNA organoids, similar to that of control organoids (Supplementary Fig. 2D). In contrast, β-catenin is relocalized throughout the cytoplasm and nucleus in the Apc-sgRNA organoids, which is characteristic of Wnt-pathway activation by Apc loss (Supplementary Fig. 2D). Taken together with the qPCR analyses, these data indicate that Cdx2 loss results in only partial activation of the Wnt pathway, and that Cdx2 does not have direct roles in suppressing Wnt-dependent transcription.

## Synergistic effects of *Sfrp4*, *Cdkn2a*, and *Sox17* loss with *Cdx2* deficiency in driving *BRAF*[V600E]-driven tumorigenesis of proximal colon stem cells

As mentioned earlier, *BRAF*[V600E]-driven colon cancers have increased likelihood of occurring in the proximal colon in the context of increased frequency of DNA methylation-mediated silencing of multiple genes, such as *Cdx2*, *Sfrp4*, *Cdnk2a*, and *Sox17*. In previous work we have demonstrated that simultaneous inactivation of *Cdx2*, *Sfrp4*, *Cdnk2a*, and *Sox17* facilitates *BRAF*[V600E]-driven tumor development in organoids derived from proximal colon[20]. As *Cdx2* inactivation in proximal and distal organoids resulted in significant growth differences, we determined the role of *Cdx2* in facilitating BRAF[V600E]-driven tumorigenesis in combination with the loss of *Sfrp4*, *Cdnk2a*, and *Sox17* in proximal and distal organoids. The data below reveal that *Cdx2* loss specifically promotes the pro-tumorigenic, Wnt-independency property in proximal but not distal colon organoids.

We performed targeted gene editing of above four genes (Cdx2, *Sfrp4*, *Cdnk2a*, and *Sox17*) simultaneously, or in combinations, using the CRISPR-AsCpf1 approach that employs linear array of guide RNAs (gRNAs) targeting these genes (Supplementary Fig. 4A)[20,38]. The suffix "gRNA" is used to term the organoid lines generated using the CRISPR-AsCpf1 approach to differentiate it from the nomenclature used for generating organoid lines using the CRISPR-Cas9 approach in the

previous section which uses single-guide RNA (sgRNA). Three combinations of these guide RNA (gRNA)-organoids were generated: C2-gRNA (targeting Cdx2), SSC-gRNA (targeting Sfrp4, Sox17, and Cdkn2a), and C2SSC-gRNA (targeting Cdx2, Sfrp4, Sox17, and Cdkn2a) (Fig. 3a, Supplementary Table 2). Sequencing the regions targeted by the guide RNAs confirmed gene editing of the targeted regions (Supplementary Fig. 4B). Wnt-independency assays on the enzymatically separated single cells with these targeted gene editing showed critical dependencies of the proximal organoids on these genes for Wnt-independent growth (Fig. 3b). Firstly, as expected, organoids with control guide RNA (Control gRNA) did not grow in WENR-minus medium with or without *BRAF*[V600E] mutation. Distal organoids were obligately dependent on Wnt-ligands for growth as none of these combinations of gene KOs (C2-gRNA, SSC-gRNA, C2SSC-gRNA) resulted in Wnt-independency, both with or without *BRAF*[V600E] activation (Fig. 3b). In contrast, proximal organoids were critically dependent on *Cdx2* mutation alone or in combination with *Sfrp4*, *Sox17* and *Cdkn2a* mutations for Wnt-independent growth, which was further enhanced by the *BRAF*[V600E] mutation (Fig. 3b). Consistent with findings in the previous section, the C2-gRNA organoids (Cdx2 loss) showed only transient Wnt-independent growth in the absence of *BRAF*[V600E] mutation, but activation of *BRAF*[V600E] led to the ability to form organoids at subsequent passages (Fig. 3b, c). In stark contrast to proximal organoids, distal organoids were not able to grow in WENR-minus medium with or without *Cdx2* mutation (C2-gRNA), and in the presence or absence of *BRAF*[V600E] mutation (Fig. 3b). The importance of *Cdx2* loss in imparting partial Wnt-independent properties is highlighted by observations that SSC-gRNA proximal organoids harboring *Sfrp4*, *Cdnk2a*, and *Sox17* mutations did not result in Wnt-independent growth regardless of *BRAF*[V600E] mutation (Fig. 3b). In contrast, targeting *Cdx2* along with *Sfrp4*, *Cdnk2a*, and *Sox17* (C2SSC-gRNA) in the proximal organoids with *BRAF*[V600E] mutation not only resulted in transient niche factor independence for the first passage (Fig. 3b), but also resulted in persistent growth at further passages (Fig. 3c). Thus, importantly *BRAF*[V600E] mutation synergized with *Cdx2*, *Sfrp4*, *Sox17*, and *Cdnk2a* mutations in promoting Wnt-independent growth only in proximal organoids. In summary, *Cdx2* loss promotes a Wnt-independent pro-tumorigenic state in stem cells of the proximal colon, which is intensified by the *BRAF*[V600E] mutation.

Subcutaneous grafting assays for tumorigenicity further shows that proximal organoids exhibited critical dependency on *Cdx2* mutation for *BRAF*[V600E]-driven tumorigenesis (Fig. 3d–f). Amongst the various groups of proximal and distal organoids injected subcutaneously into NSG mice, only C2SSC-gRNA proximal colon organoids harboring *BRAF*[V600E] formed tumors (Fig. 3d–f). *BRAF*[V600E]-activated C2-gRNA proximal organoids formed small nodular growth at the site of injection, which however did not increase in size over time (Fig. 3d). SSC-gRNA proximal organoids with activated *BRAF*[V600E] did not form tumors, thus indicating the requirement of *Cdx2* mutation for tumorigenic growth. None of the of the mutation combinations in distal organoids-initiated tumor formation, consistent with the observations above that these mutations did not impart Wnt-independent growth properties to the distal organoids. Thus, although *Cdx2* loss by itself facilitated *BRAF*[V600E]-activated proximal organoids to acquire long-term Wnt-independent growth, it was not enough to sustain subcutaneous tumors in vivo. Complete transformation of the proximal organoids in these experiments depended on combined inactivation of *Cdx2* along with *Cdkn2a*, *Sfrp4*, and *Sox17* (Fig. 3e, f, Supplementary Fig. 4C).

IHC analyses of β-catenin showed that proximal organoids with *Cdx2* loss resulted in tumors lacking canonical Wnt activation, which typically in the Apc-mutant tumors involves β-catenin translocation into the nucleus[39]. The lesions and tumors arising from C2-gRNA and C2SSC-gRNA proximal organoids, respectively, showed membranous β-catenin staining, similar to that of normal colon or normal organoids

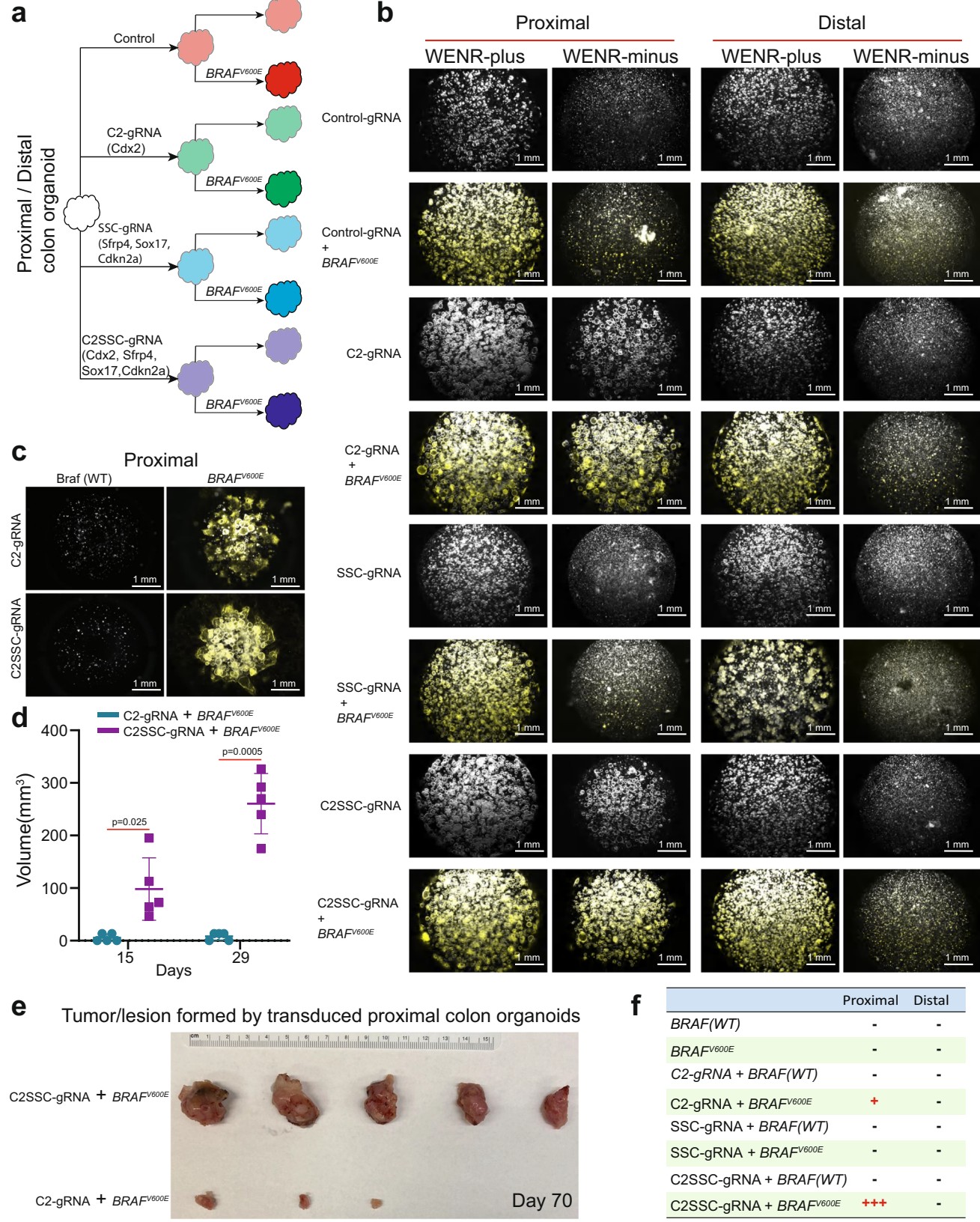

(Supplementary Figs. 2D, 4D, F). In contrast, β-catenin shows cytoplasmic and nuclear localization in tumors formed in the context of *Apc* mutation (Supplementary Fig. 4E). In general, C2SSC-gRNA tumors show loss of differentiated glandular structures and increased proliferation (Supplementary Fig. 4F). Taken together with observations from previous sections, these studies show that inactivation of *Cdx2* along with *Cdkn2a*, *Sfrp4*, and *Sox17*, which are frequently inactivated in proximal colon cancers by epigenetic silencing, synergizes with *Cdx2* loss to result in *BRAF^V600E*-driven tumor in proximal, but not distal, colon-derived stem cells.

**Fig. 3 | Mutations in *Sfrp4*, *Cdkn2a*, and *Sox17* synergize with *Cdx2* deficiency to facilitate proximal colon-specific *BRAF^V600E*-driven tumorigenesis. a** Schematic representation showing proximal and distal colon organoids transduced with lentivirus expressing control gRNA, C2-gRNA (targeting *Cdx2*), SSC-gRNA (simultaneously targeting *Sfrp4*, *Sox17*, and *Cdkn2a*), and C2SSC-gRNA (simultaneously targeting *Cdx2*, *Sfrp4*, *Sox17*, and *Cdkn2a*) for targeted mutations in these genes using CRISPR-AsCpf1 and followed with induction of *BRAF^V600E* using Cre recombinase. **b** Representative composite images (phase-contrast and red fluorescent) of Wnt-factor independency analysis for proximal and distal colon organoids treated with control Scramble-gRNA, C2-gRNA, SSC-gRNA, and C2SSC-gRNA separately with or without of *BRAF^V600E* activation (growth assay data are representative of two biological replicates). **c** Representative images showing the ability of proximal colon-derived organoids with C2-gRNA and C2SSC-gRNA to form new organoids in WENR-minus medium upon *BRAF^V600E* induction. The organoids were cultured in

WENR-minus medium for 7 days, separated into single cells, and challenged to grow in WENR-minus medium for a second passage (growth assay data are representative of two biological replicates). **d** Plot showing volume of subcutaneously grafted organoid-derived tumor/lesion formed by proximal colon organoids edited with C2-gRNA or C2SSC-gRNA combined with *BRAF^V600E* induction at 15th and 29th day after subcutaneous grafting. Error bars indicate means ± SD (*n* = 5 mice, right flank each mouse, one biological organoid replicate injected in the mice for assays). Two-sided Welch's *t*-test. Source data are provided in FigurePlotsSourceData.xlsx. **e** Representative subcutaneously grafted organoid-derived tumors dissected from the groups in (**d**) at day 70. **f** Table summarizing subcutaneously grafted organoid-derived tumor/lesion growth of proximal and distal colon organoids with control gRNA, C2-gRNA, SSC-gRNA, or C2SSC-gRNA and with or without *BRAF^V600E* induction.

## Loss of *Cdx2* results in profound gene expression changes in proximal organoids compared to distal organoids

As Cdx2 loss was the critical factor in the Wnt-independent growth and transformation of the proximal organoids, we sought to understand the underlying molecular mechanisms. We thus determined the impact of *Cdx2* loss in proximal and distal colon organoids by profiling the transcriptome (RNA-seq) of Cdx2-sgRNA organoids grown in WENR-plus and minus medium for the first passage (Supplementary Fig. 5A–C). These transcriptomic analyses reveal key differences in the transcriptional circuitry involved in stem cell and differentiation programs in the proximal and distal colon organoids. Culturing proximal and distal organoids without Wnt factors resulted in significant gene expression differences compared to culturing with Wnt factors. In the absence of exogenous Wnt signaling, 465 and 616 genes were up- and downregulated in proximal vs. distal comparisons in WENR-minus medium. In contrast, in the presence of Wnt factors, only 188 and 270 genes were up- and downregulated in proximal and distal comparisons in WENR-plus medium (Fig. 4a and Supplementary Fig. 5D). Thus, despite their similar basal transcriptomic profiles, proximal and distal colonic epithelial cells exhibit different molecular changes when external Wnt signaling is absent.

The transcriptional programs in proximal and distal organoids responded differently to Cdx2 loss in the presence or absence of exogenous Wnt signaling. Proximal organoids with Cdx2 loss exhibited significant changes in gene expression compared to control organoids when grown in either WENR-minus or WENR-plus medium, with 1404 and 1669 genes up- and downregulated in the former, and 1082 and 1236 genes up- and downregulated in the latter (Fig. 4b and Supplementary Fig. 5D). In contrast, the impact of Cdx2 loss on distal organoids was less pronounced, with smaller changes in gene expression compared to control organoids grown in either WENR-minus (467 and 536 genes up- and downregulated) or WENR-plus media (86 and 62 genes up- and downregulated) (Fig. 4b and Supplementary Fig. 5D). The above findings suggest that Cdx2 plays differential roles in regulating the transcriptional programs of proximal and distal colonic epithelial cells.

## Key growth-promoting molecular pathways are upregulated in proximal colon organoids upon loss of *Cdx2*

Geneset enrichment analyses showed that a greater number of pathways are affected upon *Cdx2* loss in proximal compared to distal organoids (Supplementary Fig. 5E). These pathways were found to be associated with key signaling pathways (Fig. 4c, d). Upregulated pathways in Cdx2-sgRNA proximal organoids grown in WENR-minus medium include proliferation (replication, cell cycle, PI3/Akt, MAPK), stem cell pluripotency, and differentiation (Hippo pathway) (Fig. 4c, d), which contrasts with the very few pathways altered in corresponding distal organoids (Fig. 4d and Supplementary Fig. 5E). These pathways are critical for tumorigenesis and matched with the organoid growth phenotype shown earlier, i.e., Cdx2-sgRNA proximal, but not

distal, colon organoids acquired transient Wnt-independent growth in the first 7-day passage. Pathways related to cell cycle and proliferation were upregulated in Cdx2-sgRNA proximal organoids, but the Wnt pathway itself was not observed to be upregulated (Supplementary dataset 1, Supplementary Fig. 5G). These changes in gene expression at the pathway level help to explain why Cdx2-sgRNA proximal organoids exhibit only transient Wnt-independent growth. However, because the Wnt pathway is not strongly activated, these organoids do not have the ability to form new organoids in WENR-minus medium.

We next investigated key biological pathways to understand the transient growth ability of *Cdx2*-deficient proximal organoids in the absence of Wnt factors during the first passage. Analyses of gene expression signatures reflecting well-defined biological processes in the Molecular Signatures Database (MSigDB) hallmark gene sets[40] showed that proliferation-related gene sets 'G2M_CHECKPOINT' and 'E2F_TARGETS' were enriched in the *Cdx2*-deficient proximal organoids compared to corresponding distal colon organoids cultured in WENR-minus medium (Supplementary Fig. 5H). These gene sets were also upregulated when Cdx2-deficient proximal, but not distal, organoids were directly compared to their corresponding control (Scramble-sgRNA) organoids cultured in WENR-minus medium (Supplementary Fig. 5I). Importantly, proliferation-related pathways are similar in *Cdx2*-deficient proximal organoids grown in WENR-plus compared to WENR-minus medium indicating that *Cdx2* deficiency sufficiently results in activation of proliferation signals in proximal organoids independent of external Wnt activation (Supplementary Fig. 5J). In contrast, in Cdx2-deficient distal organoids activation of these proliferation pathways was dependent on exogenous Wnt from WENR-plus medium, with upregulation of these pathways only in WENR-plus condition (Supplementary Fig. 5K). These data highlight that *Cdx2* loss in proximal organoids specifically renders them insensitive to absence of Wnt-ligands during the first passage in WENR-minus medium by promoting expression of biological pathways important for growth.

## Cdx2 regulates stem cell and differentiation programs specifically in the proximal colon stem cells

Based on the findings above, we asked if Cdx2 plays differential roles in maintaining colonic stem cell and differentiation phenotypes in proximal and distal colon epithelial cells. To this end we analyzed enrichment of gene signatures specific to the stem cell and differentiated cell compartments in the colon crypt. We obtained gene signatures corresponding to bottom of colon crypts (enriched for stem cells) and the top regions of the crypts (enriched for differentiated cells)[41] (Fig. 4e). Additionally, we mined gene expression signatures of human colonic epithelial cell types derived using single-cell RNA-seq analyses[42]. The crypt-bottom stem cell signature (CryptBottom_signature) is enriched in proximal organoids with *Cdx2* loss compared to corresponding distal organoids in WENR-minus medium (Fig. 4e, bottom panel). In contrast, the differentiated crypt top cell

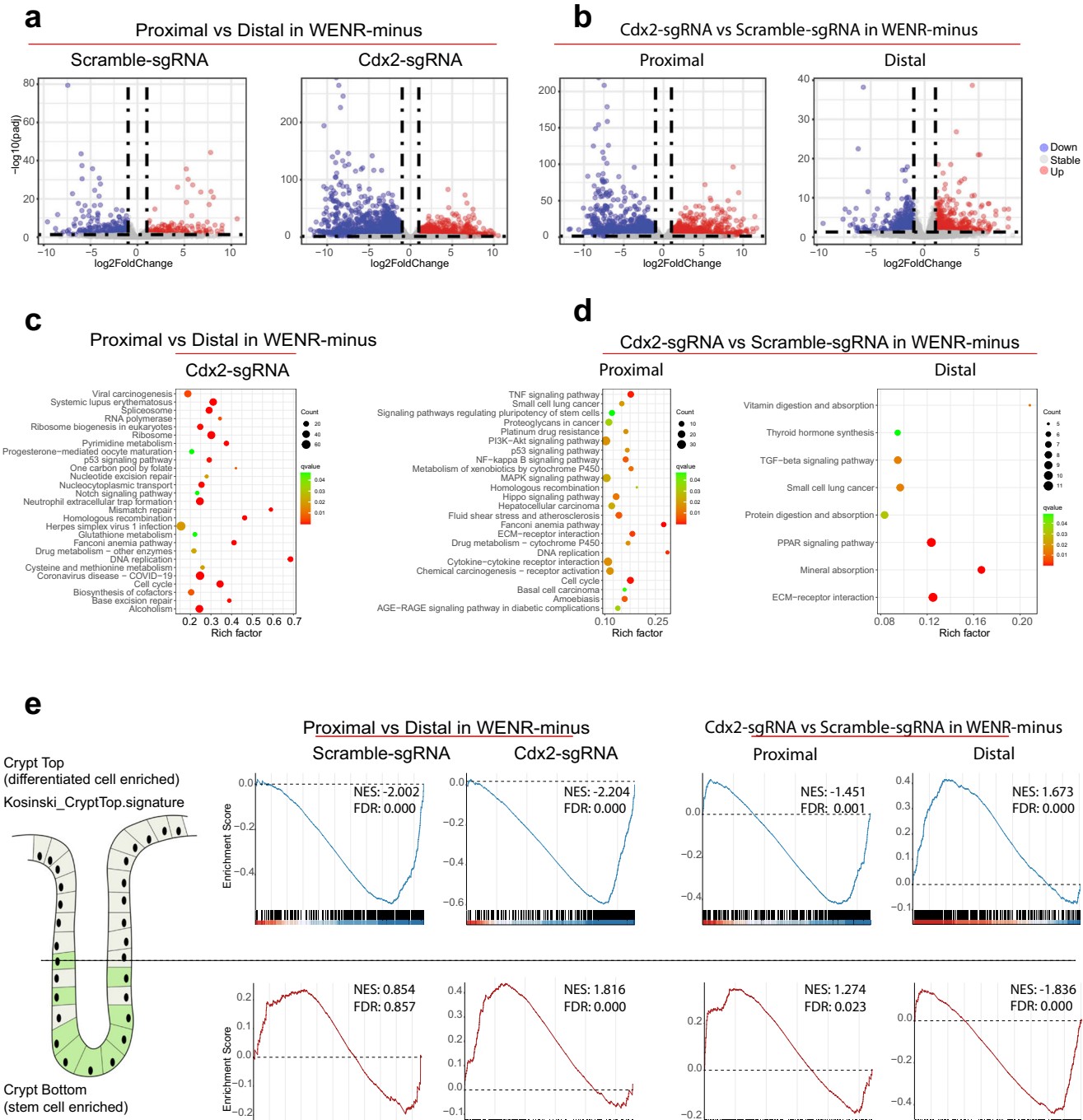

**Fig. 4 | Alterations to key pathways and stemness versus differentiation status in proximal colon organoids upon loss of *Cdx2*. a** Volcano maps representing the differentially expressed genes in proximal versus distal colon organoids, and those after *Cdx2* loss, cultured in WENR-minus medium (*n* = 2 biological replicates each). Differential gene expression analyses were corrected for multiple hypothesis using BH-methods implemented in R/DESeq2. **b** Volcano maps showing differentially expressed genes in proximal and distal colon organoids upon *Cdx2* loss compared to corresponding control organoids cultured in WENR-minus medium (*n* = 2 biological replicates each). Proximal organoids show greater number of genes with significantly altered expression compared to distal organoids. **c** Dot plots summarizing enriched KEGG pathways in the differentially expressed genes from comparing gene expression profiles of proximal versus distal colon organoids with *Cdx2* loss (Cdx2-sgRNA) cultured in WENR-minus medium (*n* = 2 biological replicates each). **d** Dot plots summarizing enriched KEGG pathways in proximal and distal colon organoids upon *Cdx2* loss compared to corresponding control organoids cultured in WENR-minus medium (*n* = 2 biological replicates each). **e** GSEA analysis showing *Cdx2* deficiency (Cdx2-sgRNA) results in decreased differentiated cell (Crypt Top) gene expression signatures and an increase in stemness (Crypt Bottom) signature in proximal colon organoids cultured in WENR-minus medium, but not distal colon organoids (*n* = 2 biological replicates each). Gene signatures are from Kosinski et al.[41].

signature (CryptTop_signature) was reduced in proximal organoids with *Cdx2* loss compared to corresponding distal organoids in WENR-minus medium (Fig. 4e, top panel). The wild-type (Scramble-sgRNA control) proximal organoids showed decrease in the differentiated cell signature compared to corresponding distal organoids in WENR-minus medium indicating inherent disposition of the proximal colon epithelial cells to reduced activation of differentiation pathways in response to absence of exogenous Wnt factors (Fig. 4e, top panel). Direct comparison of these gene expression signatures in *Cdx2*-deficient (Cdx2-sgRNA) vs. control proximal organoids in WENR-minus condition highlights the increase in stem cell signature and decrease in differentiation signature upon *Cdx2* loss specifically in the proximal organoids (Fig. 4e), while distal organoids lacking *Cdx2* showed absolutely the opposite patterns when grown in WENR-minus medium (Fig. 4e). These analyses reveal the importance of Cdx2 in regulating stem cell and differentiation related gene signatures specifically in the proximal colonic epithelial cells.

The above shifts towards increased stem and decreased differentiated states in proximal organoids upon loss of *Cdx2* were corroborated by analyses of the gene expression signatures from the single-cell datasets (Supplementary Fig. 6A–D). Gene expression signatures corresponding to differentiated cell types (colonocytes, crypt top colonocytes, enteroendocrine, goblet cells) showed a significant decrease in the Cdx2-sgRNA proximal organoids compared to corresponding distal organoids in WENR-minus medium (Supplementary Fig. 6A). The Cdx2-sgRNA proximal organoids showed reduced expression of differentiated cell markers regardless of being grown in WENR-plus or minus medium (Supplementary Fig. 6B, D). Proximal organoids with *Cdx2* loss displayed an elevated stem cell state, regardless of exogenous Wnt factors, as evident in the enrichment of the stem cell signature (Undifferentiated #1)[42] in proximal organoids when compared to distal organoids (Supplementary Fig. 6A, C). Finally, the Cdx2-sgRNA proximal, but not distal, organoids sustained increased stem cell signature independent of Wnt-ligands as the stem cell marker genes were upregulated upon Cdx2 loss regardless of being grown in WENR-plus or minus medium (Supplementary Fig. 6B, D).

Taken together, above gene expression analyses show that *Cdx2* is important in regulating cell proliferation pathways, and the stem cell and differentiation gene expression signatures in the proximal colon organoids. Activation of the proliferation and stem cell related pathways in conjunction with suppression of genes involved in differentiation in response to *Cdx2* loss in proximal organoids may thus lead to transient independence from Wnt factors, and play critical roles in facilitating *BRAF*[V600E]-induced tumorigenesis.

## Cdx2 directs transcriptional programs critical for maintenance of differentiation responsive genes in proximal colon stem cells

As Cdx2 is an important homeobox transcription factor that binds to gene regulatory elements[25], we determined its genomic targets to understand mechanisms underlying the maintenance of differential transcriptional landscape in the proximal and distal organoids by Cdx2. ChIP-seq analyses of genomic target binding sites of Cdx2 were performed in three independent mouse-derived proximal and distal colon organoid cultures grown in WENR-plus or minus medium. These analyses revealed a unique dependence of proximal organoids on Cdx2 for expression of Cdx2-bound target genes (Fig. 5a). As shown in the PCA analysis (Fig. 5b), most of the variation in the genomic binding sites of Cdx2 was derived from independent mouse groups (PC1), while the next major source of variation was derived from the proximal vs distal colon site of origin of the organoids (PC2). This latter observation is reflected in the large numbers of differential Cdx2-bound genomic sites between proximal and distal organoids, which were observed to occur independent of organoid culture in WENR-plus or minus medium (Fig. 5a).

Exogenous Wnt signaling had little impact on Cdx2 binding as the Cdx2 binding profiles of each of the proximal and distal organoids cultured in WENR-plus or minus medium tended to cluster together (Fig. 5b). Accordingly, there are very few genomic sites with differential Cdx2 binding in both proximal and distal colon organoids cultured in WENR-plus medium vs. WENR-minus medium (Fig. 5a).

Genomic region annotation of Cdx2 binding sites shows that Cdx2 binds to similar regulatory sites in both proximal and distal colon organoids. Approximately 12.5% of binding sites are located in the promoter area, while over 75% of binding sites are found within intronic and intergenic regions. These regions potentially represent enhancer elements (Fig. 5c). In total we observe 1651 number of genes with Cdx2 enrichment near the gene promoters (±3Kb around TSS) in the proximal and distal organoids across the WENR-plus and -minus media, with a significant set of genes commonly targeted among these conditions ($p$-value $< 10^{-16}$). Genes with Cdx2 enrichment at their promoters were found to be enriched for the gene expression signature of enterocyte of colon epithelium in the Tabula Muris cell types (adjusted $p$-value = 0.0005927; all other cell types, including cell types corresponding to colon stem cells did not show any significant enrichment). We addressed the differential roles of Cdx2 in gene expression regulation in proximal and distal organoid by analyzing the impact of *Cdx2* loss on expression differences of genes whose promoter is bound by Cdx2 in both proximal and distal organoids (Fig. 5d, e). Strikingly, only in the proximal colon organoids Cdx2 was found to be critical for expression of the genes whose promoter is bound by Cdx2 (Fig. 5d). Genes targeted by Cdx2 have a significant overlap with the differentially expressed genes identified in proximal organoids with Cdx2 loss compared to that if the distal organoids with Cdx2 loss in both WENR-plus and -minus media ($p$-value $\leq 0.001$, Supplementary Fig. 5L). Cdx2 loss in the proximal colon organoids resulted in significant downregulation of Cdx2-targeted genes in both WENR-plus or minus medium. In contrast, distal organoids did not show significant changes in expression of the Cdx2-targeted genes (Supplementary Fig. 5L). Importantly, the Cdx2-targeted genes were upregulated when control proximal or distal organoids are transferred to WENR-minus medium, indicating that induction of differentiation due to removal of Wnt-factors[43] is accompanied by increase in expression of these Cdx2-targeted genes (Fig. 5d). This is consistent with previous studies showing that Cdx2 controls expression of genes important in differentiation processes in colonic epithelium[25]. Our findings here show that proximal colon stem cells are more dependent on Cdx2 in regulating expression of these differentiation responsive genes.

The deep dependency of the proximal organoids on the Cdx2-maintained transcriptional program is highlighted in the geneset enrichment analyses showing significant downregulation of these genes in *Cdx2*-mutant proximal organoids compared to corresponding control organoids in both WENR-plus or minus conditions, while distal organoids are resilient to *Cdx2* loss in both WENR-plus or minus conditions (Fig. 5e). Some specific examples of the Cdx2-targeted genes upregulated in response to differentiation signals in base medium are *Car1* and *Hnf4a*, the former being a colon epithelial differentiation marker and the latter a key differentiation regulator[44,45]. This loss of upregulation with *Cdx2* deficiency for both of these genes is seen in proximal but not distal colon organoids despite the fact that their promoters are bound by Cdx2 in both proximal and distal organoids (Fig. 5f and Supplementary Fig. 5N) suggesting a model wherein proximal colon stem cells are more dependent on Cdx2 to induce a differentiation-specific transcriptional program (Fig. 5e, bottom panel). In summary, these data suggest that the suppression of the differentiation program due to Cdx2 loss may thus drive *BRAF*[V600E]-induced tumors specifically in the proximal organoids, illustrated in the previous sections.

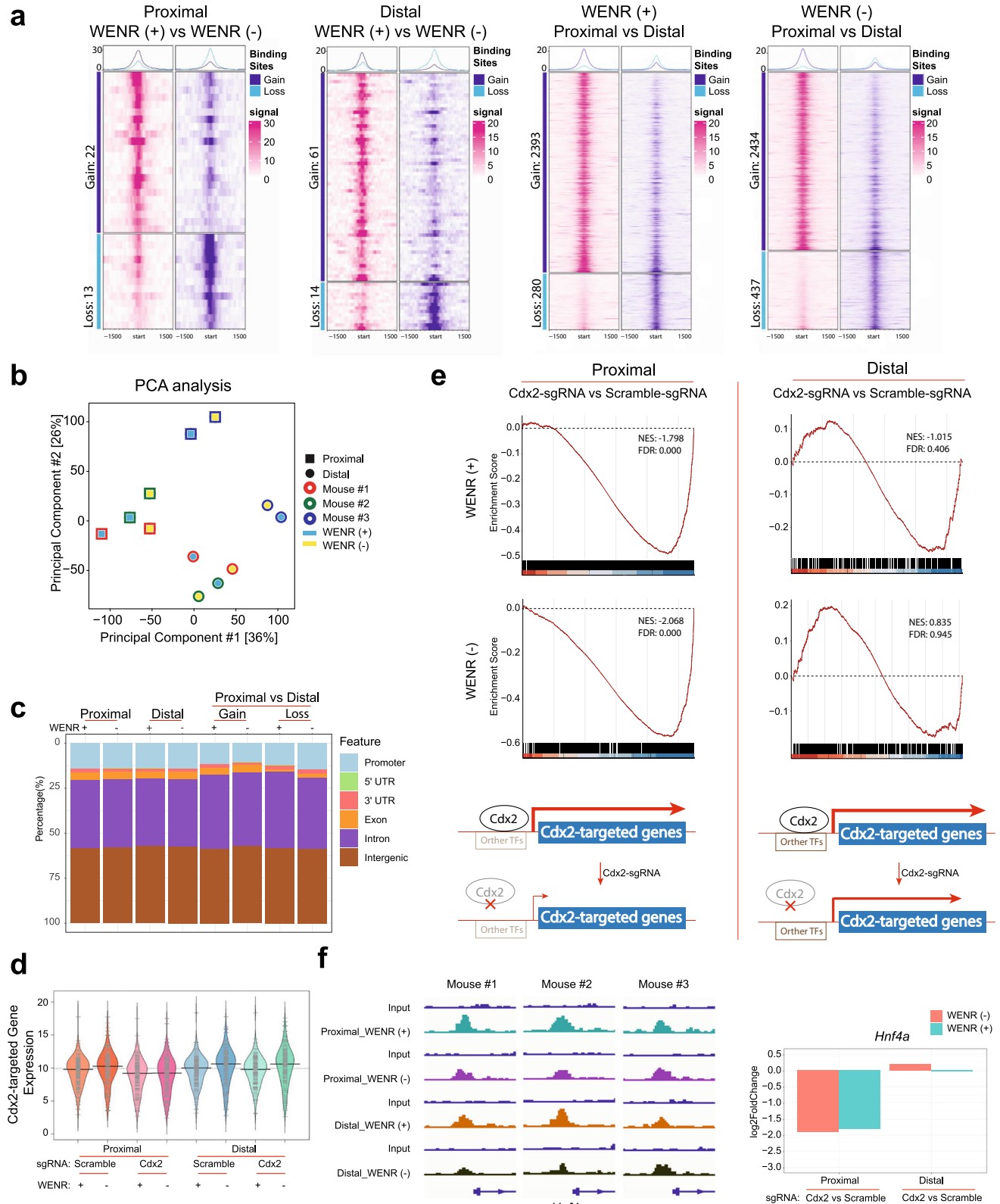

**Cdx2 loss in mouse proximal colon organoids recapitulates the gene expression patterns of human colon cancers with low *CDX2* expression**

The data in the above sections, including our mouse and in vitro organoid data, are highly correlated with the transcriptional state of CDX2 in human colon cancers. Analyses of the TCGA colon cancer data[3] shows that *CDX2* mRNA is most downregulated in proximal colon

cancers (Fig. 6a), which was further corroborated in an independent colon cancer gene expression dataset (Fig. 6b), consistent with previous findings[19]. IHC analyses in a set of proximal and distal colon cancers further show that CDX2 protein level is lower in proximal compared to distal colon cancers (Supplementary Fig. 7A–C). These associations match with observations that colon cancers with low *CDX2* expression predominantly occur in the proximal side of the

**Fig. 5 | Cdx2 directs transcriptional program critical for maintenance of differentiation responsive genes in proximal colon stem cells. a** ChIP-seq heatmaps showing differential binding of Cdx2 in proximal or distal colon organoids cultured in WENR-plus versus WENR-minus medium (the two panels on the left), and in proximal versus distal colon organoids cultured in WENR-plus or -minus medium (the two panels on the right) (*n* = 3 biological replicates). **b** PCA representation of Cdx2 enrichment at genomic loci. Each dot of the same color represents 3 replicates from independent mice-derived proximal and distal colon organoids cultured in WENR-plus or -minus medium. **c** Annotation with respect to gene structure for regions bound by Cdx2 at sites across the genome in proximal or distal colon organoids cultured in WENR-plus or -minus medium, and annotation for those regions showing gain or loss of Cdx2 binding in proximal versus distal colon organoid cultured in WENR-plus or -minus medium. **d** Violin maps summarizing the expression changes of genes whose promoters are directly bound by Cdx2 in proximal and distal colon organoids with or without Cdx2 deficiency

cultured in WENR-plus or -minus medium (*n* = 2 biological replicates for gene expression data). **e** GSEA results from comparing gene expression of genes whose promoters are bound by Cdx2 in proximal and distal organoids. Comparisons are made between Cdx2 deficient (Cdx2-sgRNA) vs. control (Scramble-sgRNA) for proximal and distal organoids grown in WENR-plus (top panel) or WENR-minus (middle panel) (*n* = 2 biological replicates for gene expression data). Schematic model diagram showing that Cdx2-bound genes in proximal organoids are dependent on Cdx2 for their expression while in distal organoids these genes are less dependent on Cdx2. **f** Left panel shows ChIP-seq enrichment traces showing Cdx2 binding in *Hnf4a* gene locus for the three replicates of proximal and distal colon organoids cultured in WENR-plus or -minus medium. Right panel shows histogram summarizing *Hnf4a* mRNA expression changes in *Cdx2* deficient (Cdx2-sgRNA) versus control proximal and distal colon organoids cultured in WENR-plus or -minus medium. Barplots depict mean log2-fold change differences in gene expression from DESeq2 analyses. (*n* = 2 biological replicates for each comparison).

colon (*p*-val < 0.001) and/or in the context of *BRAF* mutation (*p*-val < 0.001) (Fig. 6c)[19,46]. We thus tested if the transcriptional programs identified in the mouse proximal organoids lacking *Cdx2* relates to the transcriptional state in human proximal colon cancers with low *CDX2* expression. Using the TCGA colon cancer gene expression cohort, we identified gene sets whose expression is correlated with *CDX2* expression in proximal and distal colon cancers, and analyzed expression of these genes in our organoid model. The majority of these CDX2-correlated genes from human proximal colon cancers were significantly downregulated in the proximal colon organoids with *Cdx2* loss (in both WENR-minus and plus) (Fig. 6d). In contrast, CDX2-correlated genes from human distal colon cancers were not significantly altered in distal organoids upon *Cdx2* loss (Fig. 6d). Amongst the CDX2-correlated genes from human colon cancers, those downregulated in proximal organoids with *Cdx2* loss included the key epithelial regulatory genes, such as *Hnf4A*, *Satb2*, and *Vil1* (Fig. 6f). In human proximal colon cancers these genes are significantly downregulated (Fig. 6a) and furthermore their expression is significantly correlated with *CDX2* expression (Fig. 6e). Thus, disruption of *Cdx2* in proximal organoids resulted in an altered transcriptional expression state of key mediators of epithelial cell maintenance in proximal colon stem cells, such as *Hnf4a* and *Satb2*, that recapitulates the transcriptional state of human proximal colon cancers with low CDX2.

Taken together, all the studies above indicate that proximal and distal colon stem cells differ in their transcriptional states that is critical for transformation. We show that mechanistically these proximal vs. distal differences in transcriptional state are mediated by differential roles of *CDX2* in maintaining the cell differentiation programs in proximal colon stem cells. Further, *CDX2* plays an important role in regulating a wider transcriptional regulatory network specifically in proximal colon stem cells. Its loss may thus play a key mechanistic role in predisposing proximal colon stem cells to transformation by *BRAF* mutation.

## Discussion

Why tumors arising in a very closely related tissue type but different anatomical locations differ in molecular drivers of cancers is an important and basic question in cancer biology. In colon cancers, *BRAF* mutations tend to occur predominantly in the proximal colon in the context of the epigenetic CIMP-H phenotype[1,47]. Using the organoid-based colon cancer model to address the molecular dependencies for BRAF-driven colon cancers, our studies here show that transcriptional states of cells vary by the rostrocaudal axis in the colon, and importantly these transcriptional states are critical determinants of tumor development by *BRAF* mutations. Our studies here reveal *Cdx2* to be critical for maintaining stem cell and differentiation programs specifically in the proximal colon-derived epithelial cells. Mechanistically, our studies show that Cdx2 helps maintain a cell-intrinsic tumor suppressive epigenetic state specifically in proximal colon stem cells, and

that loss of *Cdx2* results in suppression of the transcriptional response to differentiation signals and partial activation of stem cell markers in the proximal colon stem cells. Importantly, Cdx2 does not directly suppress the Wnt pathway, but by being an important regulator of the differentiation program, it is required to maintain the balance between stemness and differentiation. These molecular effects of *Cdx2* loss resulted in acquisition of transient Wnt-independent growth in the proximal but not distal colon-derived organoids, which further specifically resulted in transformation of only the proximal colon stem cells by mutant *BRAF*. This dependency on *Cdx2* for maintaining stem and differentiation programs and tumor development by *BRAF* specifically in proximal colon stem cells strongly suggests that the underlying transcriptional states and its regulation play central roles in defining differences in tumors along the colonic axis.

In regard to Cdx2 being a key mediator of these transcriptional states, Cdx2 has been shown to play differential roles in embryonic and adult intestinal functions. However, its role in maintaining differential states in proximal vs. distal colon epithelium and the cancer dependency arising out of these roles has not been identified in previous studies. It is a lineage-restricted transcription factor expressed in the midgut and hindgut and is important for embryonic development of intestinal epithelium (small and large intestine)[21–28]. Embryonic loss of *Cdx2* in mice results in homeotic conversion to foregut/esophageal differentiation program[23]. In the adult intestine, Cdx2 maintains regulation of stem and differentiated cells[48], and loss of *Cdx2* leads to disruption of intestinal functions[49]. *Cdx2* is expressed at higher levels in the proximal colon of adult intestine[50], and heterozygous loss results in spontaneous intestinal adenomatous polyps, predominantly in the proximal colon[35]. Mouse models combining *Cdx2* loss with mutant *BRAF*[V600E] spontaneously harbor serrated lesions that develop into invasive carcinomas, again predominantly in the proximal colon[19]. Human colon cancers driven by *BRAF* mutation, predominantly occurring in the proximal colon, tend to occur in the context of *CDX2* downregulation by epigenetic silencing[19,20]. In a previous study, we have shown that Cdx2 loss in the context of Sfrp4, Sox17, and Cdkn2a inactivation results in complete transformation of proximal colon organoid[19]. However, it should be noted that while our earlier studies show a link between *Cdx2* loss and colon cancer development in general, the proximal vs. distal colon dependencies have remained unexplored. Our studies here using genetically engineered organoid models provide important insights into the mechanistic bases of Cdx2's differential roles in proximal vs. distal colon cancer development. We started our studies by first establishing that both proximal and distal colon organoids can be transformed by *BRAF*[V600E] and *Apc* or *Ctnnb1* mutations indicating that strong and constitutive Wnt activation sufficiently allows tumorigenesis in both proximal and distal colon stem cells. Proximal vs. distal differences in tumorigenicity were observed only when *Cdx2* was inactivated. Thus, our findings here show that *Cdx2* has starkly different roles in regulating stem and

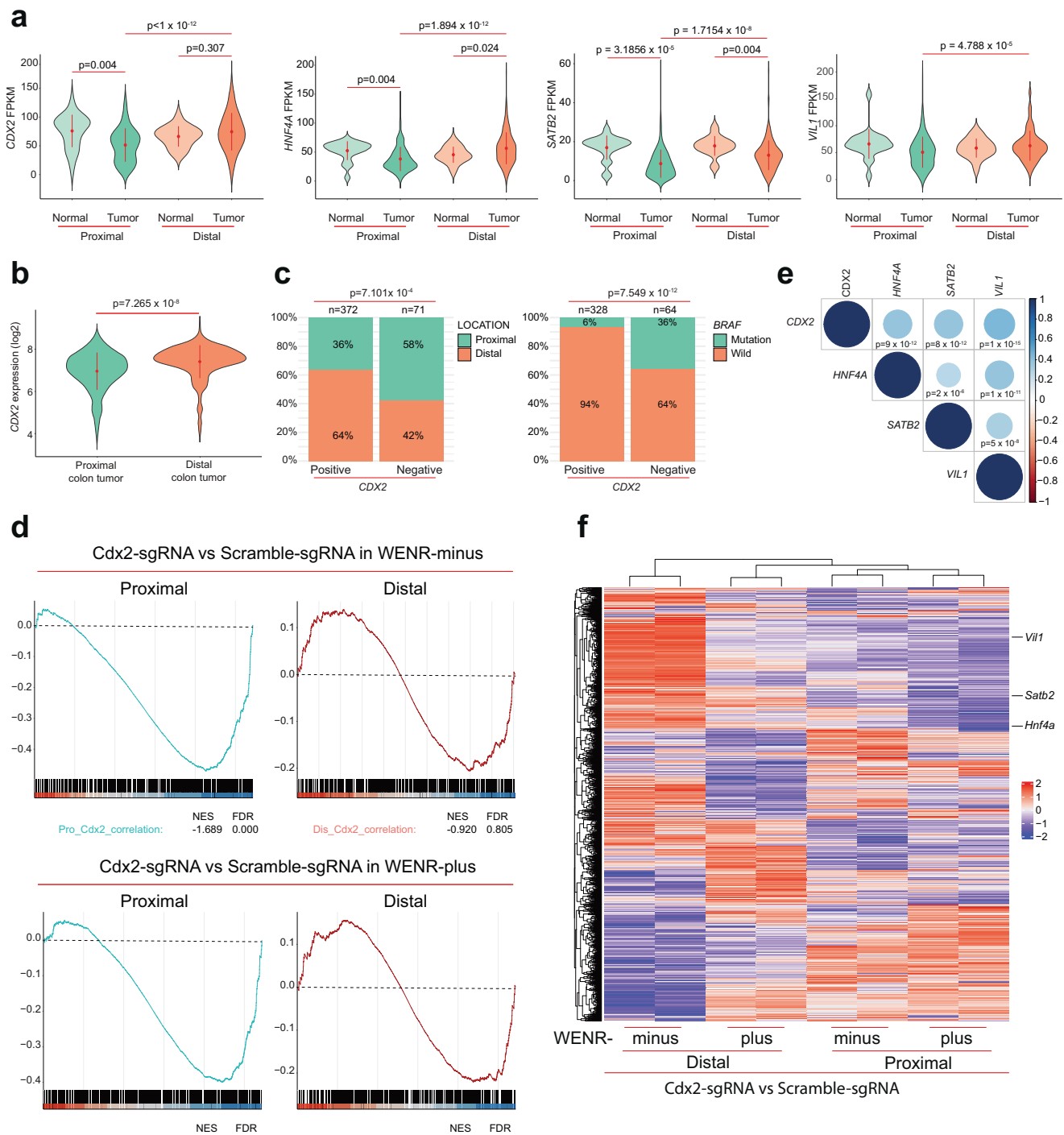

differentiation programs, and the resulting cancer dependencies, in the proximal compared to distal colon stem cells.

Our studies here have important implications for the mechanistic basis of tissue-specific molecular differences and dependencies in colon cancer development. Based on the phenotypic findings, we explored the mechanistic basis of Cdx2-mediated differential transcriptional programs in proximal and distal colon stem cells. Firstly, we observe that although Cdx2 binds to its target genes involved in differentiation in both proximal and distal colon stem cells, Cdx2 has a strong regulatory role only in the proximal colon organoid. This is potentially because Cdx2 is a primary TF regulating these target genes in proximal colon stem cells. In distal colon stem cells, where Cdx2 is basally expressed at lower levels, the Cdx2-targeted genes are

potentially subject to regulation by other transcription factors which maintain expression of these genes in the absence of Cdx2. A limitation of the current study is that the exact mechanisms underlying the differential roles of key TFs in proximal and distal colon homeostasis and tumor development are not clear. These mechanisms should be further explored to identify key TFs in distal colon and its functional relationship to Cdx2. Secondly, our ChIP-seq studies show that while Cdx2 has differential binding to target gene promoters and enhancer regulatory elements in proximal vs. distal colon stem cells, the data also suggests that Cdx2 is a key effector of a wider transcriptional regulatory network in proximal colon stem cells, such as by regulating expression of Hnf4a and Satb2, which are important differentiation regulators. The changes to pathways and cell-type gene expression

**Fig. 6 | *Cdx2* loss in mouse proximal colon organoid recapitulates gene expression patterns of human colon cancers with low *CDX2* expression. a** Violin plots showing expression of *CDX2*, *HNF4A*, *SATB2*, and *VIL1* in normal proximal colon tissue, proximal colon cancer, normal distal colon tissue and distal colon cancer in human colon cancers (TCGA-COAD) database[3] (*p*-values from one-sided Games-Howell test for post hoc analysis for comparing multiple groups with unequal variances is shown) (normal proximal colon samples = 21; proximal colon cancer samples = 293; normal distal colon samples = 20; distal colon cancer samples = 180). **b** Violin plots showing expression of *CDX2* in proximal versus distal colon cancer using GEO database (GSE39582) (*p*-values from two-sided Welch's *t*-test shown) (proximal colon cancer samples = 176; distal colon cancer samples = 267). **c** Histogram on the left shows distribution of proximal and distal tumors classified into CDX2-positive (*n* = 372) and CDX2-negative (*n* = 71) expression groups. Histogram on the right shows distribution of colon cancers with *BRAF* mutation or wild type for *BRAF* classified into the same CDX2-positive (*n* = 328) and CDX2-negative (*n* = 64) expression groups. Data were analyzed from GEO database (GSE39582)[72]. Chi-Squared test *p*-values shown. **d** GSEA results from comparing

gene expression of mouse gene orthologues of the human genes whose expression is correlated with *CDX2* expression in human colon cancers (TCGA-COAD). Comparisons are made between Cdx2 deficient (Cdx2-sgRNA) vs. control (Scramble-sgRNA) for proximal and distal organoids grown in WENR-minus (top panel) or WENR-plus (bottom panel) (*n* = 2 replicates each). The set of genes whose expression is correlated with CDX2 expression in human proximal colon cancers is used for comparing gene expression relationship in the proximal organoids (left panels), while the genes correlated with CDX2 expression in human distal colon cancers are used for comparing gene expression relationship in the distal organoid (right panels). Human colon cancer gene expression dataset is from the TCGA[3]. **e** Correlation of *CDX2*, *HNF4A*, *SATB2*, and *VIL1* gene expression in proximal colon cancer in the TCGA-COAD database[3]. *P*-values from Pearson's correlation test are shown (proximal colon cancer samples = 293). **f** Heat map showing gene expression profiles in Cdx2-KO versus Scramble-sgRNA in proximal or organoids cultured in WENR-minus or -plus medium (*n* = 2 replicates each case). Plot shows expression values for mouse gene orthologues of the human genes whose expression is correlated with *CDX2* expression in human colon cancers (TCGA-COAD)[3].

signatures show that Cdx2 loss results in a shift towards stem cell programs that is driven predominantly by suppression of differentiation signal. Multiple lines of evidence are suggestive of this: (a) tumors resulting from Cdx2 loss did not exhibit nuclear translocation of active β-Catenin; (b) gene expression and geneset enrichment analyses show that while some of the stem cell markers are upregulated, there is no overt activation of Wnt pathway; (c) When Cdx2-deficient proximal colon organoids were challenged to differentiate by removing Wnt-ligands, the differentiation gene expression program was suppressed, whereas no such effect was observed in distal colon organoids. These observations in the context of the increased expression of crypt-bottom stem cell gene signatures specifically in proximal organoids with Cdx2 loss indicates that the shift towards increased stemness in proximal colon organoids is driven by suppression of the differentiation programs. Furthermore, the transcriptional program induced by *Cdx2* loss specifically in proximal organoids was recapitulated in human proximal colon cancer samples from the Cancer Genome Atlas. We thus postulate that CDX2 potentially functions as a key effector of the tissue-location-specific epithelial cell transcriptional program by regulating a broader transcriptional regulatory network in proximal colon stem cells.

As mentioned earlier, *BRAF* mutation is predominant in proximal colon cancers in the context of CIMP-H and with a tendency to be mutually exclusive of *APC* mutations[3]. In contrast, *KRAS* mutated cancers are distributed throughout the colon tending to co-occur with *APC* mutations, with a higher tendency in the distal colon cancers[1,2]. In *BRAF*-mutant colon cancers, *CDX2* downregulation is associated with promoter DNA methylation at CpG islands, which mainly occurs in the context of CIMP-H phenotype[19]. In the etiology of these cancers, as CIMP phenotype is an early event occurring before the oncogenic *BRAF* mutation[51], our findings here suggest that downregulation of *CDX2* in the context of CIMP-H phenotype might create an early permissive state allowing *BRAF* mutation to exert its oncogenic effects. While our studies indicate how *CDX2* downregulation-driven transcriptional state may select for *BRAF* mutation in proximal colon cancers, it is intriguing what the key dependencies for *KRAS* mutation-driven tumor development are, and why *KRAS* mutations tend to occur less frequently than *BRAF* in CIMP-H cases. Most *KRAS*-mutant colon cancers tend to occur in the context of *APC* mutation, suggesting that profound Wnt activation might be a key dependency for *KRAS* mutation. We postulate that precursor mutations like *APC* and epigenetic states like CIMP define the nature of subsequent mutations driving tumorigenesis. Herein the microenvironmental factors may further be important in modulating the epigenome and influencing evolution of cancer mutations. Factors that may cause CIMP-H phenotype include an aging-like acquisition of DNA methylation, inflammation, or the microbiome microenvironment in the proximal colon[20,52]. In regard to

this, it should be noted that the in vivo dynamics of tumor initiation will involve more complexities. Using in vitro organoid cultures, here we show that the transcriptional programs in proximal and distal colon stem cells are differentially regulated by Cdx2, which drives different dependencies for tumor development. Analyses of the in vivo human colon cancer data show that the transcriptional programs in the Cdx2-deficient organoid model recapitulate those in proximal human colon cancers. Future studies using organoid models in conjunction with in vivo studies could help further address the differential dependencies for proximal vs. distal colon cancer development, as well as for *KRAS* and *BRAF* mutation-driven tumorigenesis.

In summary, our study highlights the role of developmental and lineage-restricted transcription factors, such as Cdx2, in maintaining tissue-location-specific transcriptional programs that differentially regulate stemness and differentiation, and are critical dependencies for tumor development. In this regard, a recent study has shown how positional identities of cells defined by transcriptional programs specific to cutaneous (body skin) and acral (skin at extremities) anatomic locations is a key determinant of the transforming potential of oncogenes in melanoma using a zebra fish model[53]. In relation to this study in melanoma, our studies reveal critical differences in the role of developmental programs in setting up tumor dependencies based on cancer tissue type. In the case of melanoma, these dependencies are shown to be defined by the very early body axis and limb patterning developmental programs defined by HOXB/HOX13 while in the case of colon cancers the differences in the proximal and distal colon epithelial cells arise due to differential gene expression control by later organ developmental programs regulated by the parahox CDX2. Our data provides a conceptual framework for approaching cancers occurring along different anatomical locations in the colon as distinct entities, rather than the same disease. For cancer prevention approaches, it will thus be important to consider the factors that affect changes to transcriptional programs, in addition to detecting mutations. Such dependencies on transcriptional programs may provide new opportunities for treatment approaches that takes into consideration anatomic location-specific transcriptional programs, factors affecting these programs, and the mutational drivers.

## Methods
### Contact for reagent and resource sharing
Further information and requests for resources and reagents should be directed to and will be fulfilled by Lead Contact, Hariharan Easwaran (heaswar1@jhmi.edu).

### Experimental model and subject details
**Animal models.** We used Braf^CA mice, which express normal *Braf* prior to Cre-mediated activation of a mouse-human hybrid form of

the V600E variant allele, which we have labeled as *BRAF^V600E*. Mice were bred and implemented at the Johns Hopkins Animal Care Facility. All animal protocols and care were in accordance with guidelines of the Institutional Animal Care and Use Committee (IACUC) and all experiments with mice were approved by the Johns Hopkins Animal Care and Use Committee. Briefly, bred homozygous tdTomato reporter (B6;129S6- t(ROSA)26Sortm9(CAG-tdTomato) Hze/J) mouse with heterozygous Cre-inducible *Braf^V600E* mouse(B6.129P2(Cg)-Braftm1Mmcm/J) to generate heterozygous Cre-inducible *BRAF^V600E* with heterozygous tdTomato reporter.

## Method details
Details of reagents, antibody dilutions, datasets, and software used are also provided in Supplementary Table 3.

**Research animal ethics statement.** Mice were housed at Johns Hopkins and cared for in accordance with the policies of The Johns Hopkins University Animal Care and Use Committee and our approved animal protocol. All mice used for organoid generation, subcutaneous grafting assays, or breeding were housed in rooms maintained at 67–77 °F, 30–70% RH, ventilation of 10–15 ACH, and 14:10-h light:dark cycle. Mice are housed in individually ventilated cages (JAG 75 Ventilated Cages, Allentown Caging Equipment, Allentown, PA) with 1/4" corn cob bedding (ENVIGO Teklad 1/4" Corncob Bedding, 7097, Madison, WI), a single cotton nestlet (Ancare Nestlets, NES3600, Bellmore, NY) for enrichment, and autoclavable feed (ENVIGO Teklad Global 18% Protein Extruded Rodent Diet, 2018SX, Madison, WI). Cages are autoclaved and changed every 2 weeks under a cage changing station (AG4 Animal Transfer Station, The Baker Company, Sanford, Maine). Reverse-osmosis-treated water is either hyperchlorinated or UV treated prior to distribution to cages via automated in-cage watering systems (Edstrom Industries, LLC, Waterford, WI) or water bottles (Allentown Caging Equipment, Allentown, PA). Health checks are conducted twice daily and mice are monitored quarterly for pathogens via serology and PCR of soiled-bedding sentinel mice and exhaust air duct PCR. Excluded pathogens include Sendai virus, pneumonia virus of mice (PVM), mouse hepatitis virus (MHV), minute virus of mice (MVM), mouse parvovirus 1 and 2 (MPV1, MPV2), Theiler's murine encephalomyelitis virus (TMEV, GDVII), reovirus, epizootic diarrhea of infant mice (EDIM), lymphocytic choriomeningitis virus (LCMV), ectromelia virus, murine adenovirus I and II (MAV1, MAV2, MAD), murine cytomegalovirus (MCMV), Mycoplasma pulmonis (MPUL), fur mites (Myobia, Mycoptes, Radfordia spp.), tropical rat mites (Ornithonyssus bacoti), pinworms (Aspiculuris and Syphacia spp.). Mice used for organoids generation we in addition tested to exclude for the bacterium (Helicobacter pylori) spp.

Animal breeding and harvest of tissue was performed with fewest numbers of mice necessary to reduce the numbers of animals required to satisfy the "3Rs alternatives", which refers to the replacement, reduction, and refinement of animals used in research. For monitoring in vivo tumorigenicity, -8-week-old mice with NOD.Cg-Prkdcscid Il2rgtm1Wjl/SzJ (NSG) (JAXTM background) were subcutaneously injected with the various organoid lines described further below in section "Cell counting for subcutaneous grafting model". We followed the Johns Hopkins University ACUC guidelines, which specify a maximum allowable tumor size of approximately 2 cm in any one single dimension for a single spontaneous or implanted tumor in the adult mice. This maximal tumor size/burden was not exceeded in any of the experiments. ACUC stipulates animal welfare policies as per the Animal Welfare Act regulations and Public Health Service (PHS) Policy. Accordingly, mice were monitored weekly for weight loss and other signs of distress during the course of the experiment and observed for the following symptoms of disease progression/distress which necessitates euthanizing of animal: greater than 20% loss of body weight, hypothermia, hunching, poor grooming, loss of mobility, and decreased food/water consumption.

**Organoid culture and transfection.** Female mice of age 8 weeks with heterozygous Cre-inducible *BRAF^V600E* and heterozygous *tdTomato* reporter were euthanized by Fluriso™ (VET one cat# 501017). In this work we focused on female mice to avoid gender-based variability, and because *BRAF^V600E*-driven proximal colon cancer has a higher incidence in female gender. Organoids were derived from two mice in these studies. Experiments with *BRAF^V600E* induction and *Apc*, *Ctnnb1*, and *Cdx2* knockout were done with organoids from both mice. Later studies with detailed multi-gene knockout and growth assays were done from one of the replicates. The first 1.0 cm of proximal colon and the last 1.0 cm of descending colon were extracted from the mice. Normally, organoids were plated in 6-well plate with 150 μl of Matrigel (Corning cat# 356231) and cultured in WENR-plus medium containing 50% of Wnt3a conditioned medium, 20% of R-spondin 1 conditioned medium, 10% of Noggin conditioned medium, 20% Advanced DMEM/F12 (Thermo Fisher #12634010) with B27(Thermo Fisher #17504044) and 100 ng/mL of EGF (Millipore Sigma cat# E9644-.2MG) or WENR-minus medium (without any of Wnt3a, R-spondin 1, Noggin and EGF). The conditioned media of Wnt3a, R-Spondin 1, and Noggin were made followed the protocol described previously[54]. Cre recombinase or CRISPR system was delivered into the cells using lentivirus system to active *BRAF^V600E* or for targeted gene editing, respectively. After transfection, Hygromycin (100 μg/ml) (Thermo Fisher Scientific Cat# 10687010) was used to select Cre-transduced organoids. Puromycin (2 μg/ml) (Millipore Sigma cat# P8833) or blasticidin (5 μg/ml) (Thermo Fisher Scientific cat# R21001) was used to select CRISPR-transduced organoids. Drug selection was performed for two weeks during which control organoids stopped growing.

**DNA/RNA isolation.** Each of group of organoids was embedded in 150 μl of Matrigel and cultured in 6-well plate with 2 ml of WENR-plus or WENR-minus medium. Normally, one well of organoids was used to extract DNA, two wells of organoids were used to extract RNA. The organoids embedded in Matrigel were collected using 2 ml of Cell recovery solution (Corning cat# 354253) for each well and rotated slowly in 4 °C room for 25 min. Tubes were centrifuged at 400 g (rcf) and supernatant discarded to remove dead cells on the top level of the residual Matrigel. Cells were washed one time with PBS. DNA was extracted using Quick-DNA™ miniprep kit (ZYMO research cat# D3025). RNA extracted using RNeasy Mini Kit (QIAGEN cat# 74104). The concentrations of DNA and RNA were calculated using SimpliNano (GE Healthcare).

**Cell counting for subcutaneous grafting model.** Organoids were collected using cell recovery solution, embedded in Matrigel, mixed via pipetting ten times, divided equally into 6-well plate with 150 μl Matrigel, and cultured them in WENR-plus for 6 days. After that, one well of organoid was used as a representative to count the live cell number in each well. For the well of organoids used to count the cell number, medium was discarded, Matrigel washed one time with PBS, Matrigel embedded organoids were collected using 2 ml of 5 U/μl dispase (Stemcell technologies cat# 07913) in a 5 ml tube coated with 10% FBS. Organoids were incubated at 37 °C for 10 min with intermittent rocking. After that, tube was centrifuged at 400 g (rcf) for 5 min, supernatant discarded, washed 1 time with PBS. 2 ml of Accumax was added, incubated at 37 °C for 10 min with intermittent shaking, and washed 1 time with PBS. 1 ml of PBS was added and organoids pipetted using 26-gauge syringe for 8 times. 25 μl of the cell suspension was mixed with 2× trypan blue and used to count cells. Bio-Rad's TC10 automated cell counter machine was used to count the live cell number. Cell counts from one of the wells was used as representative for the other wells in the same batch. Organoids in the remaining wells were collected similarly using cell recovery solution and pelleted. 0.75 million cells were injected subcutaneously in the right flank of NOD.Cg-Prkdcscid Il2rgtm1Wjl/SzJ (NSG) (JAXTM) with 100 μl of Matrigel. After injection,

the right flank of mice was examined every week to check the tumor formation. Tumor sizes were measured every week using digital vernier caliper and calculated using this formula $V = (L*W*H)/2$. Mice are sacrificed as soon as the tumor size reach 2000 mm³.

**pGEM-T easy vector system and Sanger sequence.** All CRISPR-transduced organoids were screened by corresponding antibiotics for 2 weeks. For each sample, 1 well of organoids was collected using cell recovery solution, pelleted by spinning, washed 1 time with PBS. After that, DNA was extract using the Quick-DNA™ miniprep kit (ZYMO research cat# D3025). PCR was used to clone the desired sequence of target gene. The PCR product was cleaned by PCR purification kit (QIAGEN cat# 28004). After that, Sanger sequencing was used to check whether there was gene mutation by Genetic Resources Core Facility at Johns Hopkins University. After checking, the residual cleaned PCR product was linked to pGEM-T Easy Vector and clone the PCR product using pGEM®-T and pGEM®-T Easy Vector Systems kit (Promega cat# A1380). After the transformed JM109 bacteria formed clone on LB/ampicillin plates spread with IPTG/X-Gal, pick up single clone into 10 ml tubes filled with 2 ml Terrific Broth (Quality Biological cat# 340-071-101) added with 10 μg/ml penicillin. Transformed *E. coli* was grown overnight at 37 °C shaking at 220 rpm. Plasmid was extracted using PureLink™ Quick plasmid Miniprep Kit (Thermo Fisher Scientific cat# K210010) and used for Sanger sequencing to confirm presence of the DNA PCR fragment clone in the T-vector.

**RT-PCR.** For each sample, 1 μg of RNA was used to generate cDNA using qScript cDNA SuperMix kit (Quantabio cat# 95048-100). In quantitative PCR, aliquots of cDNA samples with three replicates were used to detect differential expression of Fabp2, Krt20, Car1, Muc2, Ephb2, Lgr5, Ascl2 with SYBR Green technology (Bio-Rad cat# 1725124), while β-actin was applied as an endogenous control. These marker genes were used as previous studies have reported that they represent markers of stem cell or differentiated cells in the colon epithelium. The markers we used for stem cells are Ephb2, Lgr5, and Ascl2, which have been shown to be expressed at high levels in intestinal stem cells[37,55]. The differentiated cell markers used are: (a) Fabp2, which is an intestinal fatty acid-binding apoprotein expressed in intestinal enterocytes[56-58]; (b) Car1 (Carbonic anhydrase I) is expressed in colonocyte[56,57]; (c) Krt20 (Cytokeratin 20) is expressed in differentiated epithelial cells[55,59]; (d) Muc2 (mucin) is expressed in goblet cells[56,60]. The delta-delta Ct method and log2-fold change were used to calculate the relative gene expression. The exact primer sequences are listed in Supplementary Table 4. The experiment was repeated three times.

**Cloning of guide RNA in CRISPR lentivirus vector.** We use CRISPR-Cas9 gene editing technology to mutate single gene, such as *Apc*, β-catenin (*Ctnnb1*), *Sfrp4*, *Sox17*, *Cdkn2a* and *Cdx2*. To simultaneously edit several genes, we applied CRISPR-ASsCpf1 gene editing technology which could target 3 genes (SSC-gRNA: *Sfrp4*, *Sox17*, and *Cdkn2a*, simultaneously), 4 genes (C2SSC-gRNA: *Sfrp4*, *Sox17*, *Cdkn2a*, and *Cdx2*, simultaneously) as well as 1 gene (C2-gRNA: *Cdx2*). The sequence of sgRNAs or crRNAs are listed in Supplementary Table 2. The sgRNAs for Cas9 or crRNAs for AsCpf1 were designed by the CRISPR Design Tool (http://crispor.tefor.net/) and synthesized by the company of Integrated DNA Technologies (IDT). Gene-specific sgRNA oligos or crRNA oligos were cloned into lentiCRISPR v2-Blast (Addgene cat# 83480) or pY108 (lenti-AsCpf1) (Addgene cat# 84739) separately.

**Construction of Lenti-Cre and Lenti-empty vector.** We applied Gibson Assembly cloning kit (NEB cat# E5510s) to replace puromycin resistance gene in Puro.Cre empty vector (Addgene cat# 17408) by hygromycin resistance gene amplified from pBABE-hygrohTERT (Addgene cat# 1773) using primers (Hygro F and Hygro R) to generate Lenti-Cre vector. After that, primers (Cre deletion F and R) were applied to delete Cre in Lenti-Cre vector using Gibson Assembly cloning kit (New England Biolabs cat# E5510S) to generate Lenti-empty vector. All of these primers were listed in our previous study[20].

**Lentivirus packaging.** Lentivirus was packaged in Lenti-XTM 293T cells using Lipofectamine™ 3000 Transfection Reagent (Thermo Fisher Scientific cat #L3000008). Lenti-XTM 293T Cell line used were freshly obtained from Takara Bio USA, Inc., and so the cell line was not authenticated. The Lenti-XTM 293T were cultured in DMEM + 10%FBS. Cell lines were tested to be negative for Mycoplasma contamination using the MycoAlert (Lonza) kit. Contamination was tested routinely every 3–6 months. The MycoAlert by Lonza kit was used as per manufacturer's protocol. Briefly, Lenti-XTM 293T cells were cultured in T75 flask until the cells were 95% confluent. Cells were transfected with vector plasmids (Lenti-Cre vector, Lenti-empty vector, sgRNA-lentiCRISPR v2-Blast vector or gRNA-lenti-AsCpf1 vector) and packaging plasmid (psPAX2 and PMD2.G) using Lipofectamine™ 3000 Transfection Reagent. Cell culture supernatant was throwed away after 12 h and changed to 15 ml of DMEM medium with 10% FBS. After that, supernatant was collected and changed to new medium every 24 h. The collected supernatant containing lentivirus was concentrated to about 500 μl by centrifugation using Amicon® Ultra-15 Centrifugal Filter Unit (Millipore Sigma cat# UFC901024) and aliquoted to 50 μl per tube and preserved in −80 °C refrigerator before use.

**Organoid transfection.** After separation from mouse colon tissue, the proximal and distal colon organoids were cultured for 2 weeks before transfection. Briefly, organoids were dissociated using 2 ml of Accumax and incubated at 37 °C for 20 min with intermittent mixing. After washing one time with PBS, separated cells were resuspended with 200 μl of organoid culture medium containing 10 μM Y-27632 (Cell Signaling Technology cat# 13624), 10 nM CHIR99021 (Millipore Sigma cat# SML1046), 1.25 μl of TransDux (SBI System Biosciences cat# LV860A-1) and 50 μl of concentrated lentivirus particles. Drops of 200 μl of Matrigel was laid into each well of 24-wells plate and incubated at 37 °C for 20 min until Matrigel solidifies. The mixture of separated cells and lentivirus particles were overlaid on the top of the solidified Matrigel. After 16 h of incubation at 37 °C, supernatant was discarded and the cells were overlaid with fresh 200 μl of Matrigel and cultured with regular organoid culture medium. After 4–5 days, when the transfected single cells form organoids, passage these organoids into a new well of 6-wells plate, followed with 2 weeks of antibiotic selection. Organoids were tested to be negative for Mycoplasma contamination using the MycoAlert (Lonza) kit. Mycoplasma contamination was tested every 3–6 months using the MycoAlert kit as per manufacturer's protocol.

**Assessment of Wnt growth factor dependency.** For conditional medium screening, we start with two 6-well plates of organoids, each containing the same number of organoids embedded in 150 μl of Matrigel during seeding and passaging in 6-well plates. One well of organoids is used to count the cell number in culture. To do this, after culturing for 6 days, the organoid is washed using 1× PBS, and then the organoid is disintegrated to obtain single cells by using 2 ml of 5 U/μl dispase in a 5 ml tube coated with 10% FBS at 37 °C for 10 min with intermittent rocking. After that, the tube is centrifuged at 400 g (rcf) for 5 min, the supernatant is discarded, and the cells are washed once with PBS. Then, 2 ml of Accumax is added, incubated at 37 °C for 10 min with intermittent shaking, and washed once with 1× PBS. Subsequently, 1 ml of 1× PBS is added, and the organoids are pipetted using a 26-gauge syringe 8 times to completely disrupt the organoids into single cells. A 25 μl sample of the cell suspension is mixed with 2× trypan blue and used to count cells. The live cell count is performed

using Bio-Rad's TC10 automated cell counter machine. Cell counts from one of the wells serve as a representative count for the other well in the same batch. The other well of organoids designated for conditional medium screening is washed once and then lysed using 2 ml of Accumax in a 5 ml tube at 37 °C for 20 min with intermittent shaking. Finally, $1.5 \times 10^4$ cells are embedded in 10 μl of Matrigel and plated in a 24-well plate. After the Matrigel has cured, 1 ml of WENR-plus or WENR-minus medium with 10 μM Y-27632 is added. Organoids are observed under the microscope, and images are acquired on the fifth day to assess the characteristics of Wnt growth factor dependency.

For real-time PCR, organoids containing $2.5 \times 10^5$ cells embedded in 150 μl of Matrigel were loaded into the wells of a 6-well plate and cultured in WENR-plus medium for two days. In the WENR-minus treatment group, the old WENR-plus medium was removed, and the organoids were washed twice with 1× PBS. Then, 3 ml of WENR-minus medium was added, and the organoids were cultured for 3 days before being collected for RNA preparation and RT-qPCR analyses. In the WENR-plus treatment group, the organoids were continued to be cultured in fresh WENR-plus medium.

**Immunohistochemistry (IHC) and imaging.** Grafted organoid-derived tumor tissues or organoids were fixed with 4% paraformaldehyde (PFA). After dehydration, these tumor tissues or organoids were embedded in paraffin and sectioned into 2.5 μm section, followed by staining of hematoxylin and eosin (H&E). Immunohistochemistry staining for β-catenin (Cell Signaling Technology cat# 8814, 1:750), KRT20 (Cell Signaling Technology cat# 13063, 1:1000) and Ki67 (Abcam cat# ab16667, 1:200) was done by the Johns Hopkins University Oncology Tissue Services. For quantification of Kr20-positive cells in tumors from grafted organoids, five fields were chosen in every image and the total number of Krt20-positive cells counted and the proportion positive cells computed.

Organoid sizes were estimated from measuring average of the long and short diameters. The number of organoids were estimated by enumerating percentage organoids that show buds from three independent field of views.

**Western blot analysis.** Two wells of organoids cultured in 6-well plate were collected and lysed in 4% SDS. Lysates were sonicated in a water bath Bioruptor (Diagenode) for 10 min (10 cycles of 30 s on, 30 s off) at 4 °C and passed through homogenizer columns (Omega). BCA assay (Pierce Biotechnology) was applied to quantity the protein concentration. Samples were diluted using NuPAGE LDS 4x loading buffer to get 1 mg/mL concentration and boiled in boiling water for 10 min, and fractionated by SDS-PAGE using Bolt™ 4–12% Bis-Tris Plus gel (Invitrogen cat# NW04120BOX), transferred to PVDF membrane with 0.2 μm pore size (Millipore cat# ISEQ08130). Membranes were blocked with 10% Blotting-Grade Blocker (BIO-RAD cat #1706404) at room temperature for 1 h and incubated overnight at 4 °C with the following primary antibodies (diluted 1:1000): anti-CDKN2A (Abcam cat# ab211542), anti-SFRP4 (Abcam cat# ab154167), anti-CDX2 (Bethyl cat# A300-691A), anti-SOX17 (R&D cat# AF1924), anti-β-actin (CST cat# 4970S). After that, membranes were washed with TBST for 3 times and each time for 5 min, followed with the incubation for 1 h with corresponding HRP-conjugated secondary antibody: Anti-mouse IgG (CST cat# 7076), Anti-rabbit IgG (CST cat# 7074) and Anti-goat IgG (Abcam cat# ab6741). The membranes were washed with TBST for 3 times again before incubated with ECL™ Prime Western Blotting Detection Reagents (Cytiva cat# RPN2232). Finally, proteins signaling was detected by ChemiDoc™ Touch Imaging System. For the primary antibody, the dilution ratio is 1:1000, while for the second antibody, the dilution ratio is 1:2000.

**RNA-seq.** Proximal and distal colon organoids were transfected with lentivirus containing Scramble-sgRNA or Cdx2-sgRNA after they were extracted from two independent mice colon and cultured in full medium for 2 weeks. Positive transfected cells were screened using 5 μg/ml blasticidin for 2 weeks. Proximal and distal colon organoids transfected with lentivirus containing Cdx2-sgRNA were dissociated using 5U/μl dispase, followed by Acumax treatment. Separated cells were embedded in Matrigel and cultured in regular WENR-plus organoid medium in 6-wells plate. After 5–7 days, the single cell form new single-cell-originated organoid. The single-cell-originated organoid were further transferred and cultured in 12-well plate for about 6 days. Cells were collected using cell recovery solution. DNA extracted from each single-cell-originated organoids was used for determine editing of the *Cdx2* gene. Briefly, the targeted Cdx2 region was amplified by PCR, and the PCR product was cloned into T-vector plasmid. This was used to confirm editing of *Cdx2* such that it generates an out-of-frame deletion. Only single-cell-originated organoids whose two alleles containing frame shift mutation (Supplementary Fig. 5A, B), indicating complete knockout of *Cdx2* gene were used. At the fourth month of culture, different groups of organoids were split normally and cultured in WENR-plus medium for two days. Following this, one group was continued to be cultured in WENR-plus medium for three days (WENR-plus group) or in WENR-minus medium and cultured for three days (WENR-minus group). Total RNA was collected using RNeasy Plus Mini Kit (QIAGEN cat# 74104). RNA-seq libraries were prepared using SMARTer Stranded Total RNA Sample Prep Kit (Takara Bio USA cat# 634875). The pooled libraries were sequenced by Novogene Corporation INC using Hiseq Platform.

**ChIP-seq.** Proximal and distal colon organoids were generated from 3 mice and cultured in WENR-plus medium for 4 months before collecting sample for ChIP-seq. We applied Chipmentation to identify Cdx2-bound genomic regions in the organoids, with minor adaptions[61]. For WENR-plus group, organoids were cultured in WENR-plus medium, while for WENR-minus group, organoids were cultured in WENR-plus medium for 2 day first after plating in 150 μl of Matrigel, followed with 3 days treatment of WENR-minus medium. Briefly, organoids were mechanically disintegrated from the Matrigel thoroughly by pipetting, fresh 16% formaldehyde added to the suspension of 6-wells plate directly to get a final concentration of 1%. Plates were rocked on mechanical horizontal rotators for 12 min at room temperature, followed by addition of glycine at a working concentration of 0.125 M. Organoids were collected by centrifugation at 500 g (rcf) for 10 min at 4 °C. All of the subsequent experiment were done on ice and all of the buffers or solutions are ice-cold. After washing twice with 10 ml PBS containing 1 μM phenylmethyl sulfonyl fluoride (PMSF), the organoids were mixed with sonication buffer 10 mM Tris-HCl pH 8.0, 1 mM EDTA pH 8.0, 0.25% SDS, 1× protease inhibitors (Sigma-Aldrich cat#11697498001) and sonicated using a Bioruptor until most DNA fragments were between 200 and 700 bp, which was determined by agarose gel electrophoresis. The lysate was diluted at 1:1.5 ratio with equilibration buffer (10 mM Tris, 233 mM NaCl, 1.66% TritonX-100, 0.166% DOX, 1 mM EDTA, inhibitors). After spinning the sample, supernatant was transferred to a new tube, and the fragmented chromatin solution was topped up with RIPA-LS to 1000 μl. After saving 20 μl aliquot as input, 3 μg of Cdx2 antibody (Bethyl Laboratories cat# A300-691A) was added to the tube of chromatin for each Chipmentation and incubated on a rotator at 4 °C overnight. For immunoprecipitation, 15 μl protein A Dynabeads (Thermo Fisher Scientific cat# 10002D) was washed 2 times with 0.1% BSA/RIPA and incubated overnight at 4 °C to block beads. Beads were transferred to chromatin-antibody mix, followed with incubation for 2 h rotating at 4 °C. Beads were washed with RIPA-LS (twice), RIPA-HS (twice), RIPA-LiCl (twice) and 10 mM Tris pH 8, and beads were gently transferred to a new 200 μl PCR tube. Discarded the supernatant, resuspended the beads in 50 μl of tagmentation buffer containing 3 μl Tagment DNA Enzyme (Illumina cat# 20034197) and incubated at 37 °C for 16 min

with rotation. Beads were washed with RIPA-LS (twice) and TE (twice), resuspended with 48 µl ChIP elution buffer and 2 µl Proteinase K, incubate 1 h at 55 °C and 10 h at 65 °C with rotating. After reverse crosslinking formaldehyde, the DNA supernatant was purified using PCR purification kit (QIAGEN cat# 28004). To check the optimum number of enrichment cycles, RT-PCR performed as below in 10 µl reaction volume: 2 µl ChIPmentation DNA, 0.75 µM primers, 1× Kapa HiFi HotStart Ready Mix (Roche Sequencing Solutions cat# KK2601), 1× SYBR (Thermo Fisher Scientific cat# S7563), with the following program: 72 °C 5 min; 98 °C 30 s; 25 cycles of: 98 °C 10 s, 63 °C 30 s, 72 °C 30 s; 72 °C 1 min; hold at 10 °C. Calculate the Cq value of this RT-PCR, termed it as $N$. Final enrichment of libraries were made using 50 µl of PCR reaction system containing 0.75 µM primers, 1× Kapa HiFi HotStart Ready Mix, 20 µl ChIPmentation DNA with the same thermocycle program except cycles used was $N+1$. Each enriched libraries were cleaned using 1.8× volume of room temperature AMPureXP beads (Beckman Coulter cat #A63880) and using 0.60×−1.8× volume of AMPureXp beads to select 200–500 bp libraries. Finally, prepared libraries were sequenced using NovaSeq 6000 PE150 (NOVOgene).

**RNA-seq data analysis.** Sequence reads were aligned using Salmon (v1.1.0)[62] and using the gencode.vM23 version for mapping to genes. Standard alignment parameters were used by running following code: "salmon quant -i salmon_index_gencode.vM23 -l A -p 8 −1 <read1>−2 <read2>-o <output_file_name>". Transcript-level estimates were performed using R package tximport[63]. Transcript names to gene ids mappings were performed using the Bioconductor/R database TxDb.Mmusculus.UCSC.mm10.knownGene. DESeq2[64] was performed on the gene level Salmon counts data to determine differentially expressed genes for the different comparisons. Differentially expressed genes were selected after correcting for multiple hypothesis correction using the BH method. KEGG pathway and Geneset enrichment analyses (GSEA) were performed using R package clusterProfiler[65] on genes ranked based on the test statistic values from DESeq2 output.

**Chip-seq data analysis.** FastQC was utilized to certify quality of reads. Trimmomatic[66] was used to trim the ChIP-Seq reads and FastQC was used again to check whether the adapters were eliminated successfully. Bowtie2[67] was used to align reads to the mouse (mm10) genome. After marking and deleting the duplicate alignments using Samtools[68], the cleaned and mapped reads were subjected to peak calling performed by MACS2[69] peak caller with parameters 'callpeak -f BAM -g mm -B -q 0.01' and matched input was set as control. After getting the narrow peak using MACS2, ChIPseeker[70] was applied to annotate the peak calls which contain promoter, 5′UTR, 3′UTR, exon, intron and intergenic to binding and the region within ±3 Kb from transcription start site was termed as promoter. DiffBind[71] was used to find differential peaks between different group containing three biological repetitions.

For combining RNA-seq and ChIP-seq data from the organoids, the promoters that were bound by Cdx2 were identified and the expression of these genes were analyzed by plotting the distribution of the expression values and geneset enrichment analyses in the Cdx2-KO organoids grown in different medium conditions.

**Colon cancer microarray analysis of dataset from GEO.** GSE39582 dataset (https://www.ncbi.nlm.nih.gov/geo/query/acc.cgi?acc=GSE39582), the whole transcriptome arrays, containing the multi-center cohort of 443 colon cancer patients undergoing surgery from 1987 to 2007 detected by Affymetrix U133 Plus 2.0 chips were chosen for the following analysis[72]. We applied 6.5 log2 of normalized expression values of *CDX2* as the threshold, following the proposal of Dalerba et al.[73]. If *CDX2* expression ≤ 6.5, we define the tumor sample *CDX2* negative, or else positive. *T*-test was applied to check the *CDX2* expression difference between proximal colon and distal colon cancer.

Chi-Squared Test was used to analyze the correlation between *CDX2* and colon tumor location or *CDX2* and *BRAF* mutation situation.

**TCGA database analysis.** The RNA-seq data of 514 colon cancer or normal colon tissues with the corresponding clinical information were used from The Cancer Genome Atlas (TCGA). The TCGA level-3 RNA-seq data was downloaded from GDC (TCGA-COAD, lluminaHi-Seq_RNASeqV2 UNC dataset; rsem.genes.normalized_results were used; accessed Jun 14th 2018). For designating proximal and distal anatomical orientation, we classified cecum, ascending colon, and hepatic flexure as proximal colon, while we sort splenic flexure, descending colon, sigmoid colon as distal colon. It is hard to define the whole transverse colon as proximal or distal colon and distinguish whether tumor found at rectosigmoid junction is originated from distal colon or rectum, we exclude these samples for the following analysis. Genes that are positively correlated with expression of *CDX2* in proximal or distal colon cancers were obtained by setting filters of greater than 0.3 (Pearson's correlation coefficient) and Benjamini–Hochberg adjusted *p*-value ≤ 0.01.

**Colon epithelial cell-type markers from single-cell data.** Single-cell gene expression colon epithelial type markers were obtained from Parikh et al. (*Supplemental data file 41586_2019_992_MOESM3_ESM.xlsx*)[42]. Positive markers of cell clusters were obtained as genes with average logFC > 0. These represent all markers with positive gene expression identified in previous work. The Crypt Top and Bottom GeneSetList markers were obtained from Kosinski et al.[41] (Supporting information (SI) Table 1 of Kosinski et al.[41]).

**Immunohistochemistry of CDX2 in human colon cancer samples.** CDX2 expression levels in the proximal and distal colon cancers were evaluated by performing IHC (anti-CDX2, Abcam ab76541, 1:1000 dilution ratio) on a set of proximal ($n = 13$) and distal ($n = 13$) colon cancers. We quantified CDX2 protein expression in 4–5 regions of interest within each tumor sample, corresponding to tumor growth. These regions were independently identified by a pathologist. Quantification of CDX2 staining was conducted in a blinded manner using following two approaches: (A) In the first approach, we classified each tumor sample into the following groups: (a) No Expression, (b) Low Expression, (c) Intermediate Expression, (d) High Expression by visually inspecting the 4–5 regions within each tumor sample. (B) In the second approach, we calculated the ratio of the total nuclear area that tested positive for CDX2 staining to the total nuclear area (hematoxylin positive). These calculations were performed using the ImageJ plugin IHC Toolbox as described below. We trained a model to detect nuclei (hematoxylin + DAB) or only CDX2-stained nuclei (DAB) using the IHC Toolbox Plugin in ImageJ. Approximately 10 randomly selected images were used for training. These models were then utilized to identify all pixels corresponding to nuclear staining and all pixels corresponding to CDX2 staining for every region of interest marked by the pathologist. The total area in an image occupied by these identified pixels was computed through binary masking and measuring the total number of pixels. Subsequently, the ratio of total CDX2 pixels to total nuclear pixels was calculated. An ImageJ macro was used to iterate through these steps across all the regions of interest in each tumor sample. On average, five regions of interest for each patient sample were computed (total 124 regions of interest across all samples). The average of these ratios in each specimen was used as a measure of CDX2 positivity and was visualized as violin plots for proximal and distal-derived colon cancers.

**Immunohistochemistry of mouse colon.** Mice colons were processed for Swiss rolling as described by Bialkowska et al.[74]. Briefly, colonic tissues from C57BL/6 mice were cleaned with subsequent washing in ice-cold phosphate-buffered saline (PBS) twice and the entire colon

length was unraveled to free the colon from any mesenteric connective and/or fat tissue. A gavage needle attached to a 10-ml syringe filled with Bouin's fixative was inserted about half a centimeter in the anterior opening of the intestinal segment. Gentle but consistent pressure was used to flush out the contents of the intestinal segment using Bouin's fixative. This step allows simultaneous cleaning of the intestinal segment and immediate fixation (Note: Fixation can be observed by the colon color turning opaque). Using scissors, the colon was cut longitudinally and washed in ice-cold PBS twice. The cleaned and opened intestinal segment was placed in a Petri dish with the luminal side facing upward. With the help of a toothpick and forceps, the colon was rolled around the toothpick starting from the most distal portion (i.e., rectum) and with the luminal side facing upward. The distal colon was positioned in the center and the proximal colon in the outer portion of the roll resulting in a Swiss roll. Once the entire colon length was rolled up, a pair of forceps was used to carefully slide the colon Swiss roll off the toothpick and into a tissue-processing-/-embedding cassette. The cassette was placed in a container with 10% buffered formalin for 48 h before submitting it to the Histology core for further processing for immunohistochemical analysis.

## Statistics and reproducibility

No statistical method was used to predetermine sample size. No data were excluded from the analyses. The experiments were not randomized. The Investigators were not blinded to allocation during experiments and outcome assessment. All statistical details are described in the corresponding figure legends or method details section. Value of $n$ is displayed in the figure as individual data points, and in the legends. Statistical analyses were done in R (v4.2.1).

## Reporting summary

Further information on research design is available in the Nature Portfolio Reporting Summary linked to this article.

## Data availability

The RNA-seq and ChIP-seq data generated in this study have been deposited in the GEO database under accession codes GSE218480 and GSE218479, respectively, which is available at: https://www.ncbi.nlm.nih.gov/geo/query/acc.cgi?acc=GSE218482. The publicly available TCGA level-3 RNA-seq data used in this study was downloaded from GDC (TCGA-COAD, lluminaHiSeq_RNASeqV2 UNC dataset; rsem.genes.normalized_results; accessed Jun 14th 2018) and is available at: https://portal.gdc.cancer.gov/projects/TCGA-COAD. The publicly available gene expression data from 443 colon cancer patients[72] used in this study are available in the GEO database under accession GSE39582 at this hyperlink: https://www.ncbi.nlm.nih.gov/geo/query/acc.cgi?acc=GSE39582. The Crypt Top and Bottom GeneSetList markers from Kosinski et al.[41] used in our study are available at https://www.pnas.org/doi/suppl/10.1073/pnas.0707210104/suppl_file/07210table1.xls. The single-cell-based gene expression markers for colon epithelial cell types from Parikh et al. (*Supplemental data file 41586_2019_992_MOESM3_ESM.xlsx*)[42] used in this study are available at https://static-content.springer.com/esm/art%3A10.1038%2Fs41586-019-0992-y/MediaObjects/41586_2019_992_MOESM3_ESM.xlsx. The remaining data are available within the Article, Supplementary Information, or Source Data file. Source data are provided with this paper. All source data and figures are also uploaded to Figshare (https://doi.org/10.6084/m9.figshare.24850143). The Figshare data is available at https://figshare.com/s/bcda4b948ffdcf124faa. Source data are provided with this paper.

## Code availability

All software codes used in the analyses of the data deposited in GitHub (https://github.com/Baylin-Easwaran-Labs/Lijing-Yang-Project).

The Zenodo DOI for the software code submitted in GitHub is https://doi.org/10.5281/zenodo.10309481[75].

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

## Acknowledgements
Research reported in this publication was supported by: grants the National Cancer Institute under award numbers R01CA229240, R01CA230995 (to H.E., and S.B.B.,); grant from the National Institute of Environmental Health Sciences, National Institutes of Health Institute under award number R01ES011858 (to S.B.B., H.E.,); the Evelyn Grollman Glick Scholar Award (H.E.,); and the National Cancer Institute under award number P30CA006973 (SKCCC Core Grant). The content is solely the responsibility of the authors and does not necessarily represent the official views of the National Institutes of Health. We thank the Biochemistry, Cellular and Molecular Biology (BCMB) Graduate Program for graduate student support (DP). We thank Lilian Dasko-Vincent for help with microscopy (Oncology Center Support Services at the Johns Hopkins School of Medicine). We thank JoAnn Johnson, Lauren Bois and Tammy Means for administrative support. We would like to thank Jianming Zeng who teach us how to do bioinformatics analysis. We thank Laetitia Marisa, et al. for providing the GSE39582 dataset. We thank National cancer institute for providing the data of colon cancer patients.

## Author contributions
L.Y., L.T., S.B.B., and H.E. designed the study. L.Y., D.P., S.B., Y.M., and J.L. performed experiments. L.Y., L.T. analyzed/interpreted results. L.T. and S.J.T. were responsible for computational analyses. R.W.C.Y, T.L., Y.M., and N.P. provided the regents and prepared the medium for organoid culture. C.Z., K.G., and M.B. helped with mouse experiment and manuscript edition. K.J.S. supervised analyses of human colon cancer pathology specimens; L.Y., L.T., S.B.B., and H.E. wrote and edited the manuscript.

## Competing interests
The authors declare no competing interests.
