## [Peer Review File · Nature Communications]

Tissue-location specific transcription programs drive tumor dependencies in colon cancerReviewers' Comments:

Reviewer #1:

Remarks to the Author:

The current manuscript studied the difference between the proximal and distal colon tissues in tumor initiation using organoids, and found that proximal and distal colon stem cells have distinct transcriptional programs regulating stemness and differentiation. It is well known that the proximal and distal colon cells have different abilities in tumorigenesis, but the underlying mechanism remain unknown. In the study the authors proved that Cdx2 has different functions in the different parts of colons. The discovery is interesting and the data are solid. However, several questions need to be addressed before publication.

1. The authors showed that Cdx2 acts differently in the proximal and distal colon organoids. It is interesting to know whether CDX2 is important for patient tumors from different colon parts. Could the authors show the CDX2 protein expression in patient tumors from the proximal and distal colons?
2. It has been shown in the manuscript that Cdx2-loss supports transient Wnt-independent growth in proximal organoids. But the relationship between Cdx2 and WNT signaling remains unclear. Is Cdx2 required to suppress WNT-dependent transcription or Cdx2 regulates a different program?
3. Fig. 5F seems to show that WENR enhances the binding of Cdx2 in Hnf4a locus. Does it mean WNT signaling controlling Cdx2 recruitment to target genes? Also, it is not clear what the blue tracks stand for in Fig. 5F.
4. How many genes are targeted by Cdx2? Could the authors overlap the Cdx2 target genes from ChIP-Seq data with the DEGs after Cdx2 sgRNA treatment?
5. Although it has been explained in the manuscript that Cdx2 expression have a high to low gradient along the proximal to distal colon axis, it will be helpful to further clarify it to the readers by some immunostaining images of Cdx2 expression in colon tissues.
6. The results indicate that Cdx2 is not required for target gene expression regardless of its binding on them in the distal tissues. The potential mechanisms should be further discussed in the manuscript.

Reviewer #2:

Remarks to the Author:

The manuscript submitted by Yang et al., entitled "Tissue-location specific transcription programs drive tumor initiation in colon cancer" aim to describe in detail how distinct locations affect tumorigenesis and cancer initiation transcriptional programs by using a few in vivo models, in vitro 3D-organoids, CRISPR and bulk RNAseq methodology.

Some of the tools used here were generated for a previous project Tao et al, Cancer Cell 2019 as Braf mutant organoids (here from mouse model and CRISPR generation), Cdx2 KO (CRISPR generation) and Cdx2-Sfrp4-Sox17-Cdkn2a KO (CRISPR generation) combined organoids. However, it is clear that the aim of this previous project was different.

The manuscript shows some already described findings. First of all, the abstract states "it is unknown why cancers in anatomically distinct locations exhibit different molecular dependencies for tumorigenesis", which is not exactly the case. There have been many reports in this direction (by the way cited some of them in the Introduction) that can explain the different mechanisms these cells undergo. Second, the finding about Cdx2 as an important tumour suppressor gene relevant in proximal colon also has been previously defined. There are publications since the 90s and it has been studied till now due to its relevance.

In addition, there are some important limitations. The authors say that are showing here how tumour initiation is different in proximal and distal but what they show are transcriptional programs of organoids generated from single cells which strictly cannot be considered a proof of the tumour

initiation process. Indeed, one of the experiments show that Apc or Bcat KO organoids does not give rise to tumours when it has been demonstrated in many ways that aberrantly Wnt pathway activation is a tumour initiator factor in colon cancer. Furthermore, any limited dilution in vivo experiment was done to show the ability of these cells to initiate a tumour.

The novel findings are that transcriptional programs of proximal and distal organoids with Cdx2 KO or control are here better described than before.

Other concerns:

Figure 1.

Why not showing c and d in the same graph? Are there any significance when comparing proximal and distal colon tumor volume?

The panel 1e is not significant, at least there is no p-value, then why considering if it is a tendency?

Figure 2.

The novelty here is that they show each crispr ko itself and not combined. The only finding interesting is that cdx2 KO proximal organoids can survive for one passage in WENR-minus. However, when the authors try to find why only one passage and not more, the results are not explaining well the reason. for example, why Lgr5 and EphB2 in condition 6 have so different expression pattern? Here again there are no p-values or significance calculated.

Figure 3.

They use here the combination of KOs organoids and how tumours from proximal organoids are formed in xenograft experiments. This finding is linked to their previous publication, and the outcome was expected based on it.

Figure 4, 5 and 6.

Descriptive as shows the analyses of RNAseq on proximal vs distal organoids, and Cdx2 vs control.

Minor comments:

In general, I find the sections titles complex and in not all cases what is written is exactly describing the findings.

Figure 1b. The Apc or Bcat Ko organoids in WENR-plus do not show cystic phenotype as organoids derived from APC min, can this be explained?

Figure 1e. Other time-points apart of day 22 should be shown.

Figure 2c. Explain the different markers types, for example Muc2, goblet cells; Fabp2...

Figure 2c. When was this exactly done? I mean day X after passaging the organoids, or after plating the single cells? it should be written in figure legend.

Reviewer #3:

Remarks to the Author:

In the present manuscript, Lijing Yang and colleagues meticulously used both in vitro mouse organoid and in vivo xenograft models to unravel the specific mechanisms dictating tumor development dependency unique to proximal and distal colon cancer. Using a combination of ChiP-Seq, RNA-seq analyses, and existing data sets, they astutely demonstrated that the organoid model provides insight into the transcriptional variance across the rostral axis of the colon, highlighting the pivotal role of

Cdx2-mediated transcriptional programs exclusive to the proximal colon in tumor suppression. Further to this, the application of the CRISPR-Cas9 system to induce Cdx2 loss resulted in a transitory burst of Wnt-independent proliferation in organoids derived from the proximal colon. This effect, however, was not mirrored in distal colon organoid, thus underscoring the intricate region-specific response to Cdx2 loss. Moreover, it governs BRAF mutation-induced tumor development, specifically in proximal colon cancer stem cells. A further notable finding is the observed parallelism in transcriptional programs between human proximal colon cancers with down-regulated CDX2 and mouse proximal organoids lacking Cdx2. This congruence provides an invaluable link between the mouse model and human disease manifestation. The evidence compiled by the authors ultimately proposes a model where developmental transcription factors such as Cdx2 uphold tissue-specific transcriptional programs, thus establishing a tissue-type origin-specific dependence for tumor development. This model contributes significantly to understanding of the fundamental cellular mechanisms at play during the initial stages of tumor formation in colon cancer.

General comments:

In general, the reviewer believes that the findings themselves will be of interest to researchers in this field. In particular, the detailed assessment of tissue-type origin specific dependencies for tumor initiation, separately for proximal and distal colon cancer, and the revelation that the Cdx2 loss induces distinct impacts on target genes within the proximal and distal regions of the colon, achieved through the proficient use of genetically modified organoid systems, complemented by RNAseq and Chip-seq techniques, are important findings in the field of tumorigenesis. However, there are numerous problems with the experimental methods and the data presented in this manuscript. The main problem is the lack of quantitative results in organoid assays. In particular, it describes how culture conditions, tissue regions and genetic modifications affect organoid growth (Fig.1-3, S1-3), but the lack of quantitative results on organoid size, complexity, number of buds etc. makes it difficult to compare each condition.

Major Points:

1. Please provide information on detailed culture methods in organoid systems. The number of seeded cells per one Matrigel dome and the timing and method of passage are necessary to understand the role of WENR in culture and the transient Wnt-independent growth of proximal colon organoid in Figure 2.
2. Please show any quantitative results on organoid size, complexity and number of buds etc to understand differences in culture conditions. Elsewhere, the percentage of Krt20-positive cells in Fig. S2 should also be shown.
3. Cdx2 is described as a very important transcription factor in the present study, region-specific expression data for Cdx2 mRNA and protein in the respective regions of the colon used for organoid creation and in the proximal/distal organoids used in this study. In addition, the region-specific expression of Cdkn2a, Sfrp4 and Sox17 should be shown for a better understanding of Figs. 2 and S3.
4. As shown in Fig 2, the Wnt-independency assay using Cdx2 KO organoids and WENR-minus medium is very useful and can verify which parts of the colon have acquired proximal or distal-like properties. The Wnt-independency assay can be performed not only at the most proximal / distal two points but also at multiple points in between and can provide more detailed information on tissue location-specific transcriptional programs.

Minor points:

1. Please show arrows or arrowheads in "budded patterns", "vesicular structure" to facilitate understanding of the description of organoid morphology (Fig.S1b-e, S2ab).
2. There is a description to "Loss of the epigenetically silenced Cdx2 gene" in line 173, however, this is only a KO in this experiment and cannot be said to be 'epigenetically silenced'.
3. For Figure S3ef, line 205 of the text states "when grown in WENR plus or minus medium", which medium was actually used to culture the cells?
4. In Fig. S5h-k, cell proliferation-related genes were enriched in Cdx2-deficient proximal organoids, but were there any variations in cell death-related genes?
5. In Fig. 2c, "#2 scramble-sgRNA + WENR-minus" is used as a control, but in Fig. 2b, organoids are almost unobservable under those conditions. Can enough cells survive under these conditions to be

used for RNA expression analysis?

6. In Fig. S5de, please correct "WENR-minus" and "WENR-plus" in the figure as they are reversed.

Dear Reviewers,

We thank all three of the Reviewers for their time and constructive critiques. We have carefully considered each of these comments and have performed additional experiments and analyses to address them. We appreciate the suggestions, as we strongly believe they have helped clarify some points and support the key conclusions. We have extensively edited the manuscript to discuss the new data. All the changes we have made are in blue font in the revised version. We hope the reviewers will find that the changes clarify the issues that were raised.

Below, we address each of the comments and refer to the changes in the manuscript where the corresponding modifications have been made. The Reviewers' original comments are presented in black font, and our responses are indicated in blue.

REVIEWER COMMENTS

Reviewer #1, expertise in colorectal cancer epigenetics (Remarks to the Author):

The current manuscript studied the difference between the proximal and distal colon tissues in tumor initiation using organoids, and found that proximal and distal colon stem cells have distinct transcriptional programs regulating stemness and differentiation. It is well known that the proximal and distal colon cells have different abilities in tumorigenesis, but the underlying mechanism remain unknown. In the study the authors proved that Cdx2 has different functions in the different parts of colons. The discovery is **interesting** and the **data are solid**. However, several questions need to be addressed before publication.

We thank Reviewer-1 for pointing out the significance of our work and the constructive critiques. Below we provide a detailed response.

1. The authors showed that Cdx2 acts differently in the proximal and distal colon organoids. It is interesting to know whether CDX2 is important for patient tumors from different colon parts. Could the authors show the CDX2 protein expression in patient tumors from the proximal and distal colons?

To address the CDX2 protein expression levels in patient tumors, we performed immunohistochemistry (IHC) on proximal (n=14) and distal (n=12) colon cancers. These studies, which are shown below and now included in Figure S7, reveal that CDX2 expression in proximal colon cancers is lower than in distal colon cancers. We quantified CDX2 protein expression in 4-5 regions of interest within each tumor sample, corresponding to tumor growth. These regions were independently identified by a pathologist.

The quantification of CDX2 staining was conducted in a blinded manner using two approaches. In the first approach, we classified each tumor sample into the following groups: (a) No Expression, (b) Low Expression, (c) Intermediate Expression, (d) High

Expression, by visually inspecting the 4-5 regions within each tumor sample, as shown in the figure below (Figure S7).

In the second approach, we calculated the ratio of the total nuclear area that tested positive for CDX2 staining to the total nuclear area (hematoxylin positive). These calculations were performed using the ImageJ plugin IHC Toolbox.

Details of these methods are described in the resubmission (Page 52, under the section “Immunohistochemistry of CDX2 in human colon cancer samples”). The new data is included in Figure S7, and the text has been accordingly updated (Page 21, Line 21-26, “Analyses of the TCGA colon cancer”).

2. It has been shown in the manuscript that Cdx2-loss supports transient Wnt-independent growth in proximal organoids. But the relationship between Cdx2 and WNT signaling remains unclear. **Is Cdx2 required to suppress WNT-dependent transcription or Cdx2 regulates a different program?**

Our qPCR and transcriptomics data indicate that Cdx2 alters the differentiation pathways. Several lines of evidence, discussed below, indicates Cdx2 does not directly suppress Wnt-

dependent transcription, as a result its loss leads to only transient Wnt-independent growth in proximal organoids, noted by the reviewer:

(1) We observe that Cdx2-deficiency results in upregulation of the key stemness genes, such as Ephb2, Lgr5 and Ascl2, which are Wnt-pathway targets, only in the presence of external Wnt-stimulation (WENR-plus). However, unlike Apc, Cdx2 is not a direct suppressor of Wnt-target genes. As shown in Fig. 2C (below), in the absence of Wnt-ligands (WENR-minus), Lgr5 and Ascl2 are downregulated in both presence (Scramble-sgRNA + WENR-minus) or absence (Cdx2-sgRNA + WENR-minus) of Cdx2, indicating that Cdx2 does not autonomously suppress Wnt-target genes.

(2) Next, if Cdx2 is directly suppressing Wnt-dependent transcription, it should be expected that Cdx2-loss will result in upregulation of the Wnt/ β -catenin pathway. In the pathway level analyses of RNA-seq data, we found that Cdx2 deficiency does not upregulate the Wnt/ β -catenin signaling pathway in both WENR-plus or minus medium (check comparisons of Cdx2-sgRNA vs. Scramble-sgRNA; Fig. S5g; shown below).

(3) At the cellular level, Cdx2 deficiency does not change the β -catenin location in the colon cells, as shown in Fig. S4F and S2D, which is unlike Apc-loss.

4) Finally, analyses of the various cell type signatures based on the ChIP-seq and RNA-seq generated in our study indicates that Cdx2 is primarily involved in differentiation. Cdx2-loss

significantly results in downregulation of differentiated cell type signatures specifically in proximal colon organoids (Fig. 4E, Figure S6B, plot showing Cdx2-sgRNA vs Scramble-sgRNA). Furthermore, we now analyzed the Cdx2-target genes identified from the ChIP-seq data to determine cell types in which these genes are enriched. The promoter sites bound by Cdx2 are most enriched for genes expressed in the differentiated epithelial cell type of colon enterocytes (adj. pval=0.0005927), as indicated in the Mouse Tabula Muris signature shown below. Interestingly, we observe that colon stem cell signature, annotated as “epithelial cell of large intestine” in the Tabula Muris cell types, was not enriched in these analyses (adj. p-val = 0.99), indicating that Cdx2 targets genes are involved mainly in differentiated epithelial cells, and not the Wnt-target stem cell genes.

Taken together our data indicate that Cdx2 does not directly suppress the WNT-dependent transcription, and that the shift towards increased stemness in Cdx2-deficient proximal colon organoids is driven by suppression of the differentiation programs.

We have made changes in the resubmission to highlight the above points in the appropriate sections:

Results:

Page-11, Lines 22... (“It is important to note that Cdx2-deficiency...”)

Page 12, Lines 19-21 (“Taken together with the qPCR analyses...”)

Page-16, Lines 26-29 (“These changes in gene expression...”)

Page 20, Line 5-12 (“In total we observe 1651 ...”)

Discussion:

Page-23, Lines 15-17 (“Importantly, Cdx2 does not directly suppress the Wnt-pathway...”)

3. Fig. 5F seems to show that WENR enhances the binding of Cdx2 in Hnf4a locus. Does it mean WNT signaling controlling Cdx2 recruitment to target genes? Also, it is not clear what the blue tracks stand for in Fig. 5F.

The reviewer raises an interesting point that Cdx2 appears to be more enriched in the Hnf4a promoter in WENR(+) condition. We have carefully looked at various candidate genes and the overall differential ChIP-seq analyses which indicate that external Wnt-ligands do not control Cdx2 recruitment. Differential ChIP-seq analyses between WENR(+) and WENR(-) did not reveal significant numbers of differential peaks. Genome-wide, we observed only 36 and 75 differential peaks in WENR(+) vs. WENR(-) comparisons in biological replicates of proximal and distal organoids, respectively (Fig. 5a). Thus, suggesting that external Wnt-ligand does not have a strong impact on Cdx2-binding or targeting to its target genomic regions. We further looked at the Hnf4a promoter in WENR(+) vs. WENR(-) conditions. This particular region was not identified as significantly differentially enriched using the programs to identify differential ChIP-seq enrichment, indicating that the differences in binding are not significant over background. In the resubmission we show the ChIP-seq enrichment tracks for Cdx2 binding at the two key target gene promoters of Hnf4a and Car1 in each of the independent biological replicates (Figure 5F) new supplemental figure (Figure S5N). These analyses show that there is no universal trend in increase in Cdx2 recruitment to promoters in WENR (+) condition.

The blue tracks show control for ChIP-seq, which is the input DNA. We have labeled this in the updated figure.

4. How many genes are targeted by Cdx2? Could the authors overlap the Cdx2 target genes from ChIP-Seq data with the DEGs after Cdx2 sgRNA treatment?

As suggested by the reviewer, we overlapped the differentially expressed genes (DEGs) after CRISPR mediated Cdx2 knockout with the genes identified to have Cdx2 binding at their promoter region. We thank the reviewer for the suggestion as it reveals that the Cdx2-targeted genes in general are subject to gene expression changes upon Cdx2 loss in proximal but not distal colon stem cells.

In total we observe 1,651 gene-promoters (+/-3Kb around TSS) with Cdx2 enrichment in the proximal and distal organoids across the WENR-plus and -minus media, with a significant set of genes commonly targeted among these conditions (p -value $< 10^{-16}$). Genes targeted by Cdx2 are also identified to undergo significant gene expression changes upon Cdx2 loss in proximal organoids compared to distal organoids in both WENR-plus and -minus media (p -value ≤ 0.001 , Figure S5L). Importantly, genes targeted by Cdx2 get downregulated to a higher degree in the proximal organoids compared to distal organoids as revealed by negative enrichment score in GSEA analyses (Figure S5L). Thus, while Cdx2 is enriched at multiple genes in both proximal and distal organoids, its loss results in reduced expression of these Cdx2-targeted genes in proximal organoids, indicating the higher dependency of proximal colon cells on Cdx2-mediated gene expression.

Fig. S5 (I)

Updated Figure S5L shows the overlap of the DEGs with Cdx2 enriched gene targets. On the right, the heatmap showing gene set enrichment analyses highlighting that genes targeted by Cdx2 in both proximal and distal organoids are most significantly downregulated in proximal colon organoids in WENR-plus and minus conditions.

These suggestions have been incorporated on Page 20, Lines 17-23 (“**Genes targeted by Cdx2 have a significant overlap ...**”).

5. Although it has been explained in the manuscript that Cdx2 expression have a high to low gradient along the proximal to distal colon axis, it will be helpful to further clarify it to the readers by some immunostaining images of Cdx2 expression in colon tissues.

We have included a new supplemental figure (Figure S6E, F panels) where we show IHC for Cdx2 protein expression along the whole GI tract from proximal to distal colon. The figure shows that Cdx2 expression is higher in the proximal parts of the colon and significantly decreases in the distal colon.

Additionally, we have included Western blots for both proximal and distal colon in the same supplemental figure, once again demonstrating the higher expression of Cdx2 in the proximal colon.

We have also conducted further investigations into the expression of the other three proteins, Sox17, Sfrp4, and Cdkn2a, using Western blotting. The results show similar expression levels of these proteins along the colon axis, with *Cdkn2a* showing upregulation upon oncogenic stress due to *BRAF*^{V600E} induction in both proximal and distal colon organoids. The figure is shown below.

These changes can be found on page 9, lines 11-19 (“Among these genes, **CDX2** is a ...”).

6. The results indicate that Cdx2 is not required for target gene expression regardless of its binding on them in the distal tissues. The potential mechanisms should be further discussed in the manuscript.

We observe that although Cdx2 binds to its target genes in both proximal and distal colon stem cells, Cdx2 has a strong regulatory role only in the proximal colon organoid. This potentially is because Cdx2 is basally expressed at higher levels and acts as a primary TF regulating its target genes in proximal colon stem cells. In distal colon stem cells, where Cdx2 is basally expressed at lower levels, the Cdx2-targeted genes are potentially subject to regulation by other transcription factors which maintain expression of these genes in the absence of Cdx2.

Furthermore, we observe that Cdx2-loss in proximal organoids results in downregulation of Hnf4a and Satb2, which are important differentiation regulators. Thus, Cdx2 via acting as a primary regulator of various key regulators of differentiation in proximal colon stem cells, Cdx2 is potentially a key effector of a wider transcriptional regulatory network specifically in proximal colon stem cells (this was discussed in the previous submission).

We discuss that these mechanisms should be further explored to identify key TFs in distal colon and its functional relationship to Cdx2. We address the limitations of the current work in the context of the above points in the discussion (**Page-24, Line 28...**, “**Firstly, we observe that...**”).

Reviewer #2, expertise in colon cancer organoids and functional genomics (Remarks to the Author):

The manuscript submitted by Yang et al., entitled "Tissue-location specific transcription programs drive tumor initiation in colon cancer" aim to describe in detail how distinct locations affect tumorigenesis and cancer initiation transcriptional programs by using a few in vivo models, in vitro 3D-organoids, CRISPR and bulk RNAseq methodology.

Some of the tools used here were generated for a previous project Tao et al, Cancer Cell 2019 as Braf mutant organoids (here from mouse model and CRISPR generation), Cdx2 KO (CRISPR generation) and Cdx2-Sfrp4-Sox17-Cdkn2a KO (CRISPR generation) combined organoids. However, it is clear that the aim of this previous project was different.

The manuscript shows some already described findings. First of all, the abstract states "it is unknown why cancers in anatomically distinct locations exhibit different molecular dependencies for tumorigenesis", which is not exactly the case. There have been many reports in this direction (by the way cited some of them in the Introduction) that can explain the different mechanisms these cells undergo. Second, the finding about Cdx2 as an important tumour suppressor gene relevant in proximal colon also has been previously defined. There are publications since the 90s and it has been studied till now due to its relevance.

In addition, there are some important limitations. The authors say that are showing here how tumour initiation is different in proximal and distal but what they show are transcriptional programs of organoids generated from single cells which strictly cannot be considered a proof of the tumour initiation process. Indeed, one of the experiments show that Apc or Bcat KO organoids does not give rise to tumours when it has been demonstrated in many ways that aberrantly Wnt pathway activation is a tumour initiator factor in colon cancer. Furthermore, any limited dilution in vivo experiment was done to show the ability of these cells to initiate a tumour.

The novel findings are that transcriptional programs of proximal and distal organoids with Cdx2 KO or control are here better described than before.

We thank the reviewer for the insightful comments and highlighting the novelty of our finding vis à vis previous work. Regarding "it is unknown why cancers in anatomically distinct locations exhibit different molecular dependencies for tumorigenesis", this remains an actively researched area, as highlighted in the recent paper on melanoma which we have referred to (Weiss et al., Nature. 2022). In view of the reviewer's comment, in the revision we have updated the sentence to "Cancers in anatomically distinct locations exhibit different molecular dependencies for tumorigenesis". We highlight in the introduction that "What defines the mechanisms through which cancer cell-of-origin at sites along the proximal to distal (rostrocaudal) colon axis evolve different molecular dependencies for tumor initiation is not yet clarified" (Page-3, lines 19-21).

We agree with the reviewer regarding the previous studies showing that Cdx2 is an important tumor suppressor, which we have extensively referred to. However, to our knowledge this is the first such study to discover the proximal vs. distal dependencies on Cdx2, which matches to the in vivo data in mice as well as in humans. Noteworthy here are the previous studies from the Fearon group which showed that combination of BRAF mutation and Cdx2-KO in mice resulted in increased incidence of tumor initiation in proximal colon (which we have discussed in the paper). However, the proximal vs. distal specific roles of Cdx2 are not known. We hope the reviewer agrees that within this background, our studies here are novel as they now show critical mechanistic roles of Cdx2 in maintenance of differentiation and stem cell programs in the proximal and distal colon stem cells, which confers differential ability of these cells to undergo transformation/tumor initiation.

Furthermore, we hope the Reviewer agree that the organoid model is a good in vitro model system to study dependencies for tumor initiation. While the in vivo dynamics of tumor initiation will involve more complexities, the organoid model has proven to be a great in vitro system to study early genetic and epigenetic dependency for tumor initiation. The organoid model, derived from single stem cells, accordingly has been employed by many seminal studies demonstrating early roles of various cancer driver mutations and for discovery of novel cancer drivers in tumorigenesis (Drost et. al., Nature 2015; Li et. al., Nat. Med. 2014; Takeda et. al, PNAS 2019; Geurts et. al., Nat. Comm. 2023), including in our previous work showing the dependency of oncogenic mutations on epigenetic changes during early steps of tumor initiation (Tao et al., Cancer Cell. 2019). Secondly, our analyses of the in vivo human colon cancer data show that the transcriptional programs in the organoid model recapitulate those in proximal human colon cancers. We agree with the reviewer that future in vivo modeling will shed further light on the early steps in the in vivo tumor initiation process. We had briefly mentioned the need for in vivo models in the previous submission. We clarify this in the discussion of the revised manuscript (**Page 26, Lines 23...**, **“In regard to this, it should be noted that the”**).

Regarding inability of Apc-KO and β -catenin-mutant organoids to form tumors upon transplantation of the organoids in NSG mice, we would like to highlight the previous studies have similarly observed that Apc-KO organoids do not show in vivo tumorigenicity (Li et. al., Nat. Med. 2014). We observed small nodular growth at the site of injection which remains over the period of observation but did not continue to grow. In another study by the Clevers group, they showed that additional mutations in KRAS and TP53 are required along with APC for tumor formation by the organoids. The study showed that only 1/4th of the mice receiving sub-cutaneous injection of organoids carrying KRAS^{G12D}/APC^{KO}/P53^{KO} formed tumors (Drost et. al., Nature 2015). We similarly observe that loss of Apc or β -Catenin activity alone does not drive sub-cutaneous tumors, and that addition of BRAF-mutation is required for tumorigenesis in this background. Our data is thus in alignment with other studies showing that Apc-loss alone does not result in tumorigenic growth upon subcutaneous transplantation in NSG mice. We briefly discuss this point in the revised manuscript (**Page 8, Lines 8-10, “This is consistent with...”**).

Other concerns:

Figure 1.

Why not showing c and d in the same graph? Are there any significance when comparing proximal and distal colon tumor volume?

We separated 'c' and 'd' because the figure appeared overly complex, particularly for the Apc and β -catenin mutant samples. As a result, we decided to present the data separately and provide the combined summary in Figure 1e. For the reviewer's evaluation, we are displaying 'c' and 'd' together in the same plot below. However, we kindly request to maintain separate plots for the proximal and distal data.

As addressed below, there are differences in proximal and distal colon tumor volume.

The panel 1e is not significant, at least there is no p-value, then why considering if it is a tendency?

Apc-KO/Braf^{V600E}, but not β -catenin-mut/Braf^{V600E}, proximal organoid driven tumors are significantly larger than distal-organoid driven. In the revised submission, we have updated 1e (shown below). These results were discussed in **Page 8, Lines 10-12** (“When compared with distal organoids...”).

Figure 2.

The novelty here is that they show each crispr ko itself and not combined. The only finding interesting is that *cdx2* KO proximal organoids can survive for one passage in WENR-minus. However, when the authors try to find why only one passage and not more, the results are not explaining well the reason. for example, **why *Lgr5* and *EphB2* in condition 6 have so different expression pattern?** Here again there are no p-values or significance calculated.

This question pertains to the previous observations that *Ephb2* and *Lgr5* are expressed at high levels in the intestinal stem cells (Merlos-Suárez et. al. Cell Stem Cell, PMID: 21419747). It therefore is expected that these two genes should be coordinately downregulated in condition-6. We observe that in the *Cdx2*-sgRNA organoids in WENR-minus medium (condition-6, Fig. 2C), there is stark downregulation of *Lgr5* (and also *Ascl2*) but not *EphB2*. The qPCR analyses were done in the organoids after 3-days of transferring to the new medium condition (this is now detailed in the results section in the revised version, **Page11, Lines 3-8, “Organoids carrying *Apc* or *Cdx2*-sgRNA...”**). We chose 3-days for the qPCR (and RNA-seq) analyses as beyond this time point the control (reference) organoids in WENR-minus start differentiating and disintegrating. We hypothesize that the expression differences are mainly because these Wnt-target/stem cell marker genes follow different dynamics of expression within the 3-days of culturing in WENR-minus medium. It should be noted that *Cdx2*-loss influences Wnt-target genes mainly by blunting differentiation (revealed later in our study by the RNA-seq data). So, a combination of loss of external Wnt-signaling and suppression of differentiation may have different impact on expression of *EphB2* and *Lgr5/Ascl2*.

Further, as mentioned earlier (in response to Reviewer-1, Comment-2), *Cdx2*, unlike *Apc*, does not directly suppress the Wnt-target genes. Its loss did not result in strong activation

of the stem cell markers in WENR-minus. In contrast, we observe suppression of differentiation programs upon Cdx2-loss. The typical stem cell markers, like Lgr5 and Ascl2, were not upregulated in WENR-minus medium, while EphB2 levels were similar to the Scramble-sgRNA (WENR-plus) indicating that Cdx2-loss does not cause autonomous-Wnt signalling, i.e. the ability to activate Wnt-pathway in the absence of external Wnt-ligands (unlike Apc loss). However, we observe that differentiation-related markers, like Muc2, Fabp2, and Car1 are consistently downregulated in the Cdx2-KO proximal organoids in WENR-plus and minus medium. In the manuscript, we discuss that these results indicate that there is a shift in stem vs. differentiation programs upon Cdx2 loss, but not a complete activation of Wnt/ β -catenin driven stem cell program (**Discussion, Page 25, Lines 13 “The changes to pathways and...”**). In the revised version we highlight this again in the context of this figure to clarify this point better (**Results, Page-11, Lines 22..., “It is important to note that Cdx2-deficiency results ...”**). Because Cdx2 does not cause autonomous activation of stem cell programs, there is no sustained ability of these organoids to grow indefinitely in WENR-minus medium (unlike Apc-KO organoids).

In relation to the above, later when we performed the RNA-seq analyses and analyzed the gene expression programs for stem cell and differentiated cell compartments of the colon crypt, the altered stem vs. differentiation programs upon Cdx2 loss in proximal organoids is evident (Fig. 4E). These findings show that Cdx2 does not activate the canonical stem cell programs, and rather shift the balance towards retention of a stem cell program by suppressing differentiation (this was discussed in **Discussion Page 23, Line 11, “Mechanistically, our studies show”**). It should be noted that these stem cell markers (Ephb2, Lgr5 and Ascl2) are all upregulated in WENR+ medium in Cdx2-KO compared to scramble-control, indicating that in the presence of external Wnt-ligands, Cdx2-loss is associated with upregulation of the stem cell program.

We have added p-values to Figure 2c which show that the differences observed are significant. We thank the reviewer for pointing this out.

Figure 3.

They use here the combination of KOs organoids and how tumours from proximal organoids are formed in xenograft experiments. This finding is linked to their previous publication, and the outcome was expected based on it.

We agree with the reviewer that the proximal organoid phenotype is expected. However, the key question we have addressed are the roles of these genes in tumor development in proximal vs. distal colon stem cells. Going into these experiments, we did not know whether the distal organoids would behave similar to proximal organoids. As the reviewer pointed out earlier, the novelty of our current work lies in the comparison of the effects of this multi-gene knockout (KO) in proximal vs. distal colon organoids. This comparison was by design to understand the dependencies of proximal and distal colon stem cells on various genes subject to epigenetic silencing, particularly the impact of Cdx2 loss. By directly comparing the distal organoids, the analyses presented in our current work

represent a significant advance in understanding the roles of these genes in proximal and distal colon stem cells.

In agreement with the reviewer's overall assessment that the studies presented here differ from our previous work, we hope to emphasize that this study specifically focuses on the differences between the proximal and distal regions, a facet not previously studied by our group or, to the best of our knowledge, by any other group. The findings represent a substantial advancement in this field.

We have made an attempt to clarify the above points, and especially the distinction between our previous work and the current work on **Page-9, Lines 10-19, (“However, it is not known if inactivation...”)**. We hope that the significance of this study is effectively conveyed with these changes.

Figure 4, 5 and 6.

Descriptive as shows the analyses of RNAseq on proximal vs distal organoids, and Cdx2 vs control.

These figures show the RNA-seq and ChIP-Seq data that highlight functional and mechanistic differences in proximal and distal organoids upon Cdx2 loss. Further, in these figures we have integrated external datasets from crypt gene expression profiles, single cell profiles and TCGA cancer data. All together the figures show analyses underscoring our main findings that proximal and distal colon stem cells have key differences in their transcriptional circuitry, and that Cdx2 is a key factor in mediating this difference. The pathway analyses are relevant to show that Cdx2 loss causes activation of cell cycle and cancer related pathways, among alterations to various other pathways in the proximal organoids upon Cdx2 loss. We have made every attempt to present this data in the context of the key phenotypic studies, demonstrating why proximal organoids with Cdx2 loss sustain viability in the absence of Wnt factors, at least for the first passage. The figures highlight the drastic decrease in differentiated cells in the proximal organoids and a shift towards stem cell gene expression signatures and upon Cdx2 loss. We have made an attempt to present the figures in a way that highlights how these patterns are opposite in the distal organoids. Overall, these analyses help explain the different tumorigenic potential of proximal and distal colon stem cells upon Cdx2 loss and help explain the dependencies for cancer initiation.

We have made every attempt to include only most relevant data in the main figures to highlight the key differences in proximal and distal organoids; we have placed all the additional data in the supplemental figures.

In the hope of further clarifying these points so that they do not appear as a descriptive narrative, we have edited the text as below:

Page-15, Lines 18-20 (“These transcriptomic analyses reveal...”)

Page-16, Lines 14-16 (“Gene-set enrichment analyses...”)

Page-16, Lines 21-26 (“These pathways are critical for tumorigenesis...”)

Page-20, Lines 1-12 (“Genomic region annotation of Cdx2...”)

Minor comments:

In general, I find the sections titles complex and in not all cases what is written is exactly describing the findings.

We thank the reviewer for pointing out the lack of clarity in the section titles. We have carefully reviewed the titles and made changes. The titles that have been changed are in blue-colored font. We hope the updated titles provide a better description of the main findings.

Figure 1b. The Apc or Bcat Ko organoids in WENR-plus do not show cystic phenotype as organoids derived from APC min, can this be explained?

Figure 1b shows results from assaying for growth ability in WENR-plus/minus medium. In this experiment, the organoids are separated into individual cells and cultured for five days before imaging. These pictures were taken at low magnification to provide a complete field of view and to capture the degree of growth in the various conditions being tested. In the higher magnification images in Fig. S1D we show that the Apc-sgRNA organoids have cystic structures. The top panel in S1D and S1b shows that wild type organoids have budded structures. These differences are now quantified in the updated figures (Figure S1C, S3F).

Thus, Apc or β -Cat-KO organoids have the expected cystic structures. This was mentioned on **Page-7, Lines 9-11 (“Organoids with Apc or Ctnnb1...”)**.

Figure 1e. Other time-points apart of day 22 should be shown.

After day-22 the tumors in some of the conditions were too large to be continued further. Hence these mice were sacked according to the Animal Protocol Guidelines. Thus, we compared the last common time-point for all the tumors, which was day 22.

Figure 2c. Explain the different markers types, for example Muc2, goblet cells; Fabp2...

The markers we have used are now described in the methods (**Page-42, Lines 17-25, “These marker genes were used as...”**).

Figure 2c. When was this exactly done? I mean day X after passaging the organoids, or after plating the single cells? it should be written in figure legend.

We apologize for missing these details and thank the reviewer for pointing it out. We passage the organoids in a 6-well plate and culture them in WENR-plus medium for 2 days. For the WENR-minus culture group, we discard the WENR-plus medium, wash 2 times with PBS and add WENR-minus medium. For the WENR-plus culture group, we

discard the old WENR-plus medium and add fresh WENR-plus medium. After 3 days of culturing in these conditions, we collect RNA from each group of organoids.

In the revised manuscript, we have provided details in the Methods section (Page 44, Line 25, “Assessment of Wnt Growth Factor Dependency”).

In the Results section (Page 11, Lines 3-8, “Organoids carrying Apc or Cdx2-sgRNA...”), we provide a summary of the method to orient the reader.

Reviewer #3, expertise in CDX2 and GI stem cells (Remarks to the Author):

In the present manuscript, Lijing Yang and colleagues **meticulously** used both in vitro mouse organoid and in vivo xenograft models to unravel the specific mechanisms dictating tumor development dependency unique to proximal and distal colon cancer. Using a combination of ChiP-Seq, RNA-seq analyses, and existing data sets, they **astutely** demonstrated that the organoid model provides insight into the transcriptional variance across the rostral axis of the colon, highlighting the pivotal role of Cdx2-mediated transcriptional programs exclusive to the proximal colon in tumor suppression. Further to this, the application of the CRISPR-Cas9 system to induce Cdx2 loss resulted in a transitory burst of Wnt-independent proliferation in organoids derived from the proximal colon. This effect, however, was not mirrored in distal colon organoid, thus underscoring the **intricate** region-specific response to Cdx2 loss. Moreover, it governs BRAF mutation-induced tumor development, specifically in proximal colon cancer stem cells. A further notable finding is the observed **parallelism** in transcriptional programs between human proximal colon cancers with down-regulated CDX2 and mouse proximal organoids lacking Cdx2. This congruence provides an **invaluable** link between the mouse model and human disease manifestation. The evidence compiled by the authors ultimately proposes a model where developmental transcription factors such as Cdx2 uphold tissue-specific transcriptional programs, thus establishing a tissue-type origin-specific dependence for tumor development. This **model contributes significantly to understanding** of the fundamental cellular mechanisms at play during the initial stages of tumor formation in colon cancer.

General comments:

In general, the reviewer believes that the findings themselves will be of interest to researchers in this field. In particular, the detailed assessment of tissue-type origin specific dependencies for tumor initiation, separately for proximal and distal colon cancer, and the revelation that the Cdx2 loss induces distinct impacts on target genes within the proximal and distal regions of the colon, achieved through the **proficient** use of genetically modified organoid systems, complemented by RNAseq and Chip-seq techniques, are **important findings** in the field of tumorigenesis. However, there are **numerous problems with the experimental methods** and the data presented in this manuscript. The main problem is the lack of quantitative results in organoid assays. In particular, it describes how culture conditions, tissue regions and genetic modifications affect organoid growth (Fig.1-3, S1-3), but the lack of quantitative results on organoid size, complexity, number of buds etc. makes it difficult to compare each condition.

We thank Reviewer-3 for the several positive comments. Below we address the various concerns raised by the reviewer.

Major Points:

1. Please provide information on detailed culture methods in organoid systems. The number of seeded cells per one Matrigel dome and the timing and method of passage are necessary to understand the role of WENR in culture and the transient Wnt-independent growth of proximal colon organoid in Figure 2.

In the revision, we add these details in the Methods section (**Page 44, Line 25, "Assessment of Wnt Growth Factor Dependency"**) as below:

For conditional medium screening, we start with two 6-well plates of organoids, each containing the same number of organoids embedded in 150 μ l of Matrigel during seeding and passaging in 6-well plates. One well of organoids is used to count the cell number in culture. To do this, after culturing for 6 days, the organoid is washed using 1x PBS, and then the organoid is disintegrated to obtain single cells by using 2 ml of 5 U/ μ l dispase in a 5 ml tube coated with 10% FBS at 37°C for 10 minutes with intermittent rocking. After that, the tube is centrifuged at 400 g (rcf) for 5 minutes, the supernatant is discarded, and the cells are washed once with PBS. Then, 2 ml of Accumax is added, incubated at 37°C for 10 minutes with intermittent shaking, and washed once with 1x PBS. Subsequently, 1 ml of 1x PBS is added, and the organoids are pipetted using a 26-gauge syringe 8 times to completely disrupt the organoids into single cells. A 25 μ l sample of the cell suspension is mixed with 2X trypan blue and used to count cells. The live cell count is performed using Bio-Rad's TC10 automated cell counter machine. Cell counts from one of the wells serve as a representative count for the other well in the same batch. The other well of organoids designated for conditional medium screening is washed once and then lysed using 2 ml of Accumax in a 5 ml tube at 37°C for 20 minutes with intermittent shaking. Finally, 1.5×10^4 cells are embedded in 10 μ l of Matrigel and plated in a 24-well plate. After the Matrigel has cured, 1 ml of WENR-plus or WENR-minus medium with 10 μ M Y-27632 is added. Organoids are observed under the microscope, and images are acquired on the fifth day to assess the characteristics of Wnt growth factor dependency.

For real-time PCR (Fig. 2C), organoids containing 2.5×10^5 cells embedded in 150 μ l of Matrigel were loaded into the wells of a 6-well plate and cultured in WENR-plus medium for two days. In the WENR-minus treatment group, the old WENR-plus medium was removed, and the organoids were washed twice with 1x PBS. Then, 3 ml of WENR-minus medium was added, and the organoids were cultured for 3 days before being collected for RNA preparation and RT-qPCR analyses. In the WENR-plus treatment group, the organoids were continued to be cultured in fresh WENR-plus medium.

2. Please show any quantitative results on organoid size, complexity and number of buds etc to understand differences in culture conditions. Elsewhere, the percentage of Krt20-positive cells in Fig. S2 should also be shown.

We now show the size of the organoids in Figure S1C and S3F; the number of organoids with buds as a measure of the complexity is shown in the images as percent cells with budded structures. The quantification highlights that in general the proximal organoids tended to grow into larger structures with small buds while the distal organoids were smaller with more budded protrusions. Induction of *BRAF^{V600E}* is associated with larger organoid sizes, indicative of increased proliferation (quantification in Figure S1C, shown below).

Secondly, *Cdx2* loss in proximal organoids results in increased organoid size compared to distal organoids (quantification in Figure S3F, shown below), as well as loss of budded structures compared to the distal organoids (images in Figure SE).

Figure S1C. Dot plot showing the size (long plus short diameter divided by 2) of proximal and distal colon organoids edited by Apc-sgRNA or β -Catenin-sgRNA, with or without induction of $BRAF^{V600E}$. Two-sided Wilcoxon rank sum exact test.

Figure S3F. Dot plot showing the size (long plus short diameter divided by 2) of proximal and distal colon organoids edited by Cdx2-sgRNA. P-values calculated using two-sided Wilcoxon rank sum exact test.

We have quantitated Krt20 positive cells in Figure S2 (shown below). The data indicates a tendency for increase in Krt20 positive cells in distal organoids derived tumors with Apc loss, which however did not pass statistical significant thresholds. These differences for β -Catenin-loss tumors were not significant. We observe that there are more clusters of Krt20 positive cells in the Apc and β -Catenin-loss tumors, as observed in Figure S2B. We have updated the text to discuss these results (Page-8, Lines 18-26, “**Further, BRAFV600E driven tumors in the context...**”).

3. Cdx2 is described as a very important transcription factor in the present study, region-specific expression data for Cdx2 mRNA and protein in the respective regions of the colon used for organoid creation and in the proximal/distal organoids used in this study. In addition, the region-specific expression of Cdkn2a, Sfrp4 and Sox17 should be shown for a better understanding of Figs. 2 and S3.

As mentioned earlier in response to Reviewer-1, and here to Reviewer-3, we have included Cdx2 expression by IHC and western blot in proximal and distal parts of the colon in the revised submission (**Figure S6E, F**). These analyses show that Cdx2 is expressed at higher levels in proximal organoids.

Furthermore, we have included expression of Cdk2na, Sfrp4 and Sox17, which are expressed to similar levels in proximal and distal colon. Cdkn2a is specifically upregulated upon $Braf^{V600E}$ expression, in response to the oncogenic stress.

4. As shown in Fig 2, the Wnt-independency assay using Cdx2 KO organoids and WENR-minus medium is very useful and can verify which parts of the colon have acquired proximal or distal-like properties. The Wnt-independency assay can be performed not only at the most proximal / distal two points but also at multiple points in between and can provide more detailed information on tissue location-specific transcriptional programs.

We completely agree with the reviewer that testing the effects of Cdx2-KO on Wnt-independent growth in different parts of the colon will be a crucial experiment. In our current analyses, we focused on the extreme ends of the rostro-caudal axis to examine key differences and found some intriguing results. In principle, this should be followed by detailed molecular studies of the organoids from various parts of the colon. While it is a great suggestion, for the sake of timely dissemination of the findings in this study, we hope to conduct similar experiments in different parts of the colon in future studies.

Minor points:

1. Please show arrows or arrowheads in “budded patterns”, “vesicular structure” to facilitate understanding of the description of organoid morphology (Fig.S1b-e, S2ab).

We have updated Figure S1 and S2 to show arrows to point to the relevant morphologies highlighted in the Results section. The legend indicates the morphological features as suggested by the reviewer.

2. There is a description to “Loss of the epigenetically silenced Cdx2 gene” in line 173, however, this is only a KO in this experiment and cannot be said to be 'epigenetically silenced'.

It was meant to highlight that Cdx2 is normally lost by epigenetic silencing. We see the reviewers point that this can cause confusion. Accordingly, we have modified the title to “**Loss of Cdx2 gene imparts transient Wnt-independent growth specifically in proximal colon organoids**”

3. For Figure S3ef, line 205 of the text states “when grown in WENR plus or minus medium”, which medium was actually used to culture the cells?

We apologize for the confusion. WENR-plus medium was used to culture the cells in In Figure S3E and F. This has been corrected (**Page 10, Lines 2, “when grown in WENR-plus medium**).

4. In Fig. S5h-k, cell proliferation-related genes were enriched in Cdx2-deficient proximal organoids, but were there any variations in cell death-related genes?

We have analyzed for cell death-related apoptosis pathways in the GSEA analyses of the Hallmarks pathways. These pathways were not enriched in Cdx2-deficient organoids in the GSEA analysis.

5. In Fig. 2c, “#2 scramble-sgRNA + WENR-minus” is used as a control, but in Fig. 2b, organoids are almost unobservable under those conditions. Can enough cells survive under these conditions to be used for RNA expression analysis?

This relates to this Reviewer’s question-1 above, and also to one of Reviewer-2’s question (“other concerns” relating to figure 2), about details of the methods used for generation of the organoids for the images in Fig. 2b and the RNA for the qPCR data in Fig 2c. In Figure 2b, organoids were imaged for conditional medium screening. In Figure 2c, the organoids were processed for RNA analyses. The qPCR analyses were done in the organoids after 3-days of transferring to the new medium condition (this is now detailed in the results section in the revised version, **Page11, Lines 3-8, “Organoids carrying Apc or Cdx2-sgRNA were...”**). We chose 3-days for the qPCR (and RNA-seq) analyses as beyond

this time point the control (reference) organoids in WENR-minus start differentiating and disintegrating.

Also, these details are in the Methods section (**Page 44, Line 25, “Assessment of Wnt Growth Factor Dependency”**) as described above to this Reviewer’s question-1.

6. In Fig. S5de, please correct “WENR-minus” and “WENR-plus” in the figure as they are reversed.

We thank the reviewer for pointing out this error, which has been fixed in the revised submission.

Reviewers' Comments:

Reviewer #1:

Remarks to the Author:

The authors have nicely addressed all my concerns. I do not have any more questions.

Reviewer #2:

Remarks to the Author:

The manuscript has been improved and Cdx2 relevance in proximal versus distal colon is better supported than in the previous version. However the novelty is still the same as in the previous version. No major findings have been added. Cdx2 relevance has been largely explained by previous manuscripts and here they identify that its locations has an impact in tumorigenesis.

I think some of the responses are vague and not answer totally my main concerns. In addition, some data of figures as 1c,d and e, are part of them no significant so should not be considered as results.

In the end, they reveal some mechanistic insights of cdx2 transcriptional program comparing two localizations of the colon which affect tumorigenesis. I don't agree these type of experiments in organoids can be extrapolated to declare tumour initiation abilities (written in the title).

I also don't think that the title should be written as they had study all the complex transcriptional programs of the two locations which is not the case. They have studied Cdx2 impact by RNAseq and Chip.

Reviewer #3:

Remarks to the Author:

It is of great satisfaction that the authors have thoughtfully addressed the concerns and suggestions raised in the initial review.

Furthermore, the additional data strengthens the authors' findings, and this revised manuscript demonstrates their commitment to quality research.

Dear Reviewers,

We thank all three of the Reviewers again for their careful consideration of the manuscript and the various comments which has immensely helped improve the manuscript.

Below, are some specific comments to the 2nd revision. The Reviewers' original comments are presented in black font, and our responses are indicated in blue.

REVIEWER COMMENTS (after 1st revision)

REVIEWERS' COMMENTS

Reviewer #1 (Remarks to the Author):

The authors have nicely addressed all my concerns. I do not have any more questions.

We again appreciate Reviewer-1 for the constructive comments that has helped improve the manuscript.

Reviewer #2 (Remarks to the Author):

The manuscript has been improved and Cdx2 relevance in proximal versus distal colon is better supported than in the previous version. However the novelty is still the same as in the previous version. No major findings have been added. Cdx2 relevance has been largely explained by previous manuscripts and here they identify that its locations has an impact in tumorigenesis.

I think some of the responses are vague and not answer totally my main concerns. In addition, some data of figures as 1c,d and e, are part of them no significant so should not be considered as results.

In the end, they reveal some mechanistic insights of cdx2 transcriptional program comparing two localizations of the colon which affect tumorigenesis. I don't agree these type of experiments in organoids can be extrapolated to declare tumour initiation abilities (written in the title).

I also don't think that the title should be written as they had study all the complex transcriptional programs of the two locations which is not the case. They have studied Cdx2 impact by RNAseq and Chip.

We thank Reviewer-2 again for the various positive comments. We agree with the various concerns raised which are important aspects to address in further work.

We have modified the title to remove the "initiation" term. Below is the modified title:

“Tissue-location specific transcription programs drive tumor dependencies in colon cancer”

Reviewer #3 (Remarks to the Author):

It is of great satisfaction that the authors have thoughtfully addressed the concerns and suggestions raised in the initial review.

Furthermore, the additional data strengthens the authors' findings, and this revised manuscript demonstrates their commitment to quality research.

We again are grateful to Reviewer-3 for the various constructive comments.